



# AGCPP: All-day Global Cloud Physical Properties dataset with 0.07° resolution retrieved from geostationary satellite imagers covering the period from 2000 to 2022

Lingxiao Zhao[1], Feng Zhang[1], Jingwei Li[1], Feng Lu[2], and Zhijun Zhao[1]

[1]Key Laboratory of Polar Atmosphere-Ocean-Ice System for Weather and Climate of Ministry of Education / Shanghai Key Laboratory of Ocean-Land-Atmosphere Boundary Dynamics and Climate Change, Department of Atmospheric and Oceanic Sciences & Institutes of Atmospheric Sciences, Fudan University, Shanghai, China
[2]CMA Key Laboratory for Cloud Physics, Weather Modification Center, China Meteorological Administration (CMA), Beijing, China

*Correspondence to*: Feng Zhang (fengzhang@fudan.edu.cn)

**Abstract.** The use of remote sensing to accurately measure cloud properties and their spatial and temporal variability has become an important area of atmospheric science research. However, the heterogeneity of data formats across national agencies and the calibrate and navigate associated with the use of data from different agencies have prevented the climate research community from using the full continuum of global cloud physical properties products. In this paper, All-day Global Cloud Physical Properties (AGCPP) is proposed, which provides cloud physical properties covering nearly the entire globe, from latitude -70° to 70° and longitude -180° to 180°. The main attributes of this dataset include cloud phase, cloud top height, cloud optical thickness, and cloud effective radius, with a time range from 1 January 2000 to 31 December 2022. AGCPP combines the observational advantages of geostationary satellites and polar-orbiting satellites. It uses the Moderate Resolution Imaging Spectroradiometer (MODIS) Level-2 cloud product (MOD06/MYD06) to train the cloud-based attention-UNet (CloudAtUNet) model, and then evaluates AGCPP using MOD06/MYD06 and the Cloud–Aerosol Lidar with Orthogonal Polarisation (CALIOP) 1 km cloud layer product. The evaluation results indicate that AGCPP demonstrates excellent continuity and consistency in both temporal and spatial accuracy, as well as high consistency in diurnal accuracy. Due to the long time series and all-day global nature of the dataset, it is expected that the dataset AGCPP will significantly increase the potential for climate change research, particularly with respect to potential feedback effects between clouds, surface albedo, and radiation. AGCPP is stored in the Network Common Data Format (netCDF), a standard that allows various tools and libraries to process the data quickly and easily. The AGCPP dataset is freely available on the Science Data Bank at https://doi.org/10.57760/sciencedb.26292 (Zhao et al., 2025), and the corresponding code can be found at https://github.com/lingxiao-zhao/AGCPP (last access: 25 June 2025).



# 1 Introduction

Cloud processes occupy a central role throughout the lifecycle of both severe convective storms and tropical cyclones. From their initial formation and growth to eventual dissipation, the evolution of cloud microphysics exerts a direct control on precipitation intensity and its spatial distribution (Zhuge and Zou, 2018; Yan et al., 2024). Likewise, the structural evolution and intensity fluctuations of tropical cyclones are intimately tied to ice–water phase transitions and the size distribution of cloud droplets within convective cores (Zhuge et al., 2015; Hsieh et al., 2024). Beyond modulating rainfall patterns, cloud layers also regulate Earth's radiation budget by scattering incoming shortwave radiation and absorbing outgoing longwave radiation (Liu et al., 2024b; Viggiano et al., 2025), and they play a pivotal role in the global hydrological cycle (Liu et al., 2024b; Viggiano et al., 2025). Consequently, acquiring high-quality, global-scale observations of cloud physical properties—and resolving their spatiotemporal variability—remains indispensable for advancing both weather forecasting and climate-system research.

Currently, polar-orbiting satellite sensors, such as the Aqua and Terra (Platnick et al., 2015) satellites equipped with the Moderate Resolution Imaging Spectroradiometer (MODIS), have been continuously providing high-spatial-resolution (approximately 1 km) physical property datasets for global cloud cover since 2000. Compared to geostationary satellites operating in geosynchronous orbits, polar-orbiting satellites, due to their low Earth orbits of approximately 700 km, can obtain more accurate brightness temperature and cloud property retrieval data (Frey et al., 2008). These data are widely used in studying the interactions between clouds and climate change (Brennan et al., 2005; Kaps et al., 2023). For example, MODIS's cloud mask algorithm supports multiple spectral bands (day and night compatible) and has been verified through radar/lidar experiments to have higher accuracy than the Advanced Very High Resolution Radiometer (AVHRR) (Liu et al., 2004)). However, polar-orbiting satellites only scan along the Earth's poles in narrow bands (approximately 2,000 km wide), making it impossible to achieve continuous, comprehensive observations of the global cloud field (Menzel et al., 2008). In contrast, geostationary satellites (orbiting at an altitude of approximately 36,000 km) can continuously monitor approximately one-third of the Earth's surface day and night, providing high-frequency observations at minute intervals for long-term cloud changes.

However, due to the higher orbital altitude of geostationary satellites, the accuracy of the brightness temperature and cloud physical property retrievals obtained is slightly inferior to that of polar orbiting satellites (Zhang et al., 2021). Additionally, the sensors on geostationary satellites from various countries each have their own limitations: The Advanced Himawari Imager (AHI) on Japan's Himawari-8 (Wang et al., 2024) and the Meteosat Second Generation/Spinning Enhanced Visible and Infra-Red Imager (MSG/SEVIRI) on European Organisation for the Exploitation of Meteorological Satellites (EUMETSAT) (Poulsen et al., 2012; Watts et al., 2011) only release daytime cloud physical property products using the Optimal Cloud Analysis (OCA) algorithm, lacking nighttime products; China's Fengyun series (FY-4A/B) Advanced Geostationary Radiation Imager (AGRI) has also not yet released nighttime cloud microphysical parameter products including cloud optical thickness and cloud effective radius (Chen et al., 2020; Zhou et al., 2024); the U.S. GOES



16/17 series Advanced Baseline Imager (ABI) provides cloud microphysical parameters covering both daytime and nighttime, but due to the use of two separate Cloud Optical and Microphysical Properties (COMP) retrieval algorithms for daytime and nighttime (Walther and Heidinger, 2012), the algorithm switch causes discontinuous jumps or biases in the product during day-night transitions, introducing false signals in long-term climate trend analyses at timescales longer than a day (Smalley and Lebsock, 2023).

The limitations of using geostationary satellite data to derive cloud physical products are primarily due to the shortcomings of traditional retrieval algorithms. Firstly, daytime retrievals primarily rely on the visible and shortwave infrared dual-spectral bands (e.g., DCOMP and OCA algorithms). These methods can only accurately estimate cloud optical thickness and particle size under conditions with solar radiation, but they often fail or experience a significant drop in accuracy during twilight, dawn, and nighttime conditions (Wolters et al., 2008). Second, while traditional nighttime methods (such as Optimal Estimation (Iwabuchi et al., 2014) or infrared split-window (Heidinger and Pavolonis, 2009)) can be used at night, they are limited by the penetration capability of thermal infrared radiation, leading to systematic biases in estimating the microphysical properties of thick clouds or highly reflective cloud layers (Mayer et al., 2024).

In recent years, advances in high-performance computing and artificial intelligence have promoted the application of machine learning and deep learning methods in the field of cloud property retrieval. Pérez et al. (2009) first used neural networks to retrieve the MODIS infrared radiation model to support nighttime microphysical processes. Subsequent studies further constructed a unified day-night retrieval model based on cross-channel feature learning to improve retrieval accuracy and efficiency (Lee et al., 2021; Kurihana et al., 2022; Kotarba and Wojciechowska, 2025; Gao et al., 2024), particularly achieving significant improvements under thick cloud conditions (Zhao et al., 2023; Min et al., 2020). However, these methods are only applicable to sensors with similar orbits. Additionally, since each geostationary satellite only covers a regional area and stores data in fragmented, heterogeneous archives, creating a global continuous cloud layer attribute product poses significant challenges in data collection and pre-processing. For example, calibrating raw sensor signals to radiant brightness (Helder et al., 2020; Lee et al., 2024) and brightness temperature, as well as navigation to map each pixel to Earth surface coordinates (Knapp et al., 2011; Jiao et al., 2024). To address this issue, some studies have begun to integrate polar orbiting satellite data with geostationary satellite observations (Tong et al., 2023; Li et al., 2023; Zhao et al., 2024; Liu et al., 2025). The Gridded Satellite (GridSat-B1) project pioneered a truly global, spatio-temporally continuous brightness temperature dataset by stitching together infrared channel data from multiple geostationary satellites (Knapp et al., 2011), laying the foundation for seamless, long-term climate analysis products (Shi et al., 2025; Letu et al., 2023; Tang et al., 2025).

The objective of this study was to combine the high spatio-temporal resolution brightness temperature observations of geostationary satellites with the high-precision advantages of polar-orbiting satellite cloud products to construct the All-day Global Cloud Physical Properties (AGCPP).First, this study constructed a large-scale training sample by matching the infrared brightness temperature of GridSat-B1 and ERA5 meteorological fields with the high-resolution cloud physical products of the MODIS satellite at the pixel level. Then, a deep neural network was trained to learn the mapping relationship.



Finally, the model weights obtained from the training were applied to the entire GridSat-B1 brightness temperature time series to produce the AGCPP dataset. The innovation of this study lies in the fact that AGCPP is the first global dataset with a spatial resolution of 0.07° and a temporal resolution of 3 h. As shown in Table 1, this is the latest cloud product dataset
currently available worldwide. Firstly FY-4B cloud products, CARE, GOES-R ABI cloud products, Himawari-8 cloud products are regional cloud products while AGCPP is a global cloud product. Secondly International Satellite Cloud Climatology Project (ISCCP) and CLARA-A3 have coarser spatial resolution of 0.3° and 0.25° and AGCPP has 0.07°. Finally, SatCORPS Global Cloud Product starts only from 2023 and is not capable to do long time climate analysis, while AGCPP covers 23 years from 2000-2022. Due to the long-term time series and all-day global characteristics of AGCPP
dataset, it is anticipated that the dataset will significantly enhance the potential for climate change research, particularly studies on the potential feedback effects between clouds, surface albedo, and radiation.

**Table 1.** Comparison results between our dataset and the latest cloud physical property product dataset.

| Agencies | Names | Region | Latitude | Longitude | Products | Spatial resolution | Time resolution | Years |
|---|---|---|---|---|---|---|---|---|
| Ours | AGCPP | Global | 70° S–70° N | 180° W-180° E | cloud phase, cloud top height, cloud optical thickness, cloud effective radius | 0.07° | 3 h | 2000-2022 |
| National Satellite Meteorological Centre, China Meteorological Administration (NSMC-CMA) (Zhang et al., 2024) | FY-4B cloud products | China/Full Disc | 80° S–80° N | 23.8° E-173.8° W | cloud phase, cloud top height, cloud optical thickness (daytime), cloud effective radius (daytime) | 0.036 °(macro)/ 0.018 °(micro) | 0.25 h (macro)/1 h (micro) | 2018-present |
| Aerospace Information Research Institute, Chinese Academy of Sciences (AIRI-CAS) (Letu et al., 2020) | CARE | East Asia–Pacific region | 10° S–60° N | 60° E-180° E | cloud phase, cloud top height, cloud optical thickness (daytime), cloud effective radius (daytime) | 0.1° | 0.5 h | 2016-present |
| National Oceanic and Atmospheric Administration (NOAA) (Heidinger et al., 2020) | GOES-R ABI cloud products | Western Hemisphere (Full Disc, CONUS, mesoscale) | 80° S–80° N | 142° E-56° W/156° W-6° E | cloud phase, cloud top height, cloud optical thickness and cloud effective radius | 0.018° | 5–15 min (FD), 5 min (CONUS), 30–60 s (Mesoscale) | 2018–present |





| | | | | | | | | |
|---|---|---|---|---|---|---|---|---|
| Japan Aerospace Exploration Agency (JAXA) (Mouri, 2019) | Himawari-8 cloud products | East Asia–Pacific region | 60° S–60° N | 70° E-150° W | cloud phase, cloud top height, cloud optical thickness (daytime), cloud effective radius (daytime) | 0.045° | 0.25 h (10 min) | 2015-present |
| National Aeronautics and Space Administration (NASA) (Young et al., 2018) | ISCCP (DX, D1, C1, H-series) | Global | 90° S–90° N | 180° W–180° E | cloud phase, cloud top height, cloud optical thickness | DX: 0.3°, C1/D1: 2.5°, H: 1° | 3 h (DX, D1, C1), 1 month (H) | DX/D1/C1: 1983–2009, H: 1983–2017 |
| EUMETSAT (Karlsson et al., 2023b; Karlsson et al., 2023a) | CLARA-A3 | Global | 90° S-90° N | 180° W-180° E | cloud phase, cloud top height, cloud optical thickness, cloud effective radius (daytime) | 0.25° | 24 h/ 1 month | 1979-2020 |
| NASA (Minnis et al., 2008; Minnis et al., 2021) | SatCORPS Global Cloud Product | Global | 90° S-90° N | 180° W-180° E | cloud phase, cloud top height, cloud optical thickness (daytime), cloud effective radius (daytime) | 0.027° | 1h | 2023-present |

This paper provides further details on the AGCPP dataset records, including input data, algorithm explanations, product examples, and validation results. Section 2 briefly introduces data preparation and methods, while Section 3 introduces, discusses and evaluation the four main product groups: cloud phase (CLP), cloud top height (CTH), cloud optical thickness (COT), and cloud effective radius (CER). Section 4 presents the basic characteristics of AGCPP. Section 5 describes the availability of the data. Section 6 is the conclusion.


## 2 Data and methods

### 2.1 Data

#### 2.1.1. Geostationary satellite data

Since the 1970s, geostationary satellites have been regularly providing high-temporal-resolution Earth observation data.
However, conducting climate research using their extensive historical data has typically faced significant obstacles. Key challenges include the absence of a global central repository integrating all international satellite data, the difficulty of processing massive amounts of spatiotemporal resolution data, and the heterogeneity of calibration and navigation formats across different satellite systems. These factors have added complexity to the unified processing required for multi-satellite climate research. To overcome these limitations, institutions such as the National Oceanic and Atmospheric Administration

(NOAA) National Climatic Data Center (now NCEI) have remapped data to standard projections, recalibrated them to improve temporal consistency, and ultimately created datasets such as GridSat-B1 (Knapp et al., 2011).

The primary source of the geostationary infrared channel brightness temperature data used in this study is the GridSat-B1 dataset. Over the 23-year study period from 2000 to 2022, a total of 24 satellites from four countries or regions participated in the construction of the GridSat-B1 dataset, as shown in Fig. 1. This global dataset integrates observational

data from multiple geostationary satellites to provide top-of-atmosphere (TOA) infrared brightness temperature (from two bands at infrared water vapor (IRWVP) 6.7 μm and infrared window (IRWIN) 11 μm). Specifically, for each grid point, the measurement closest to the satellite's nadir is selected. GridSat-B1 features a standard grid resolution of 0.07° (8 km) and a temporal resolution of 3 hours, corresponding to standard weather observation times of 0000, 0300, ..., 2100 UTC. Image acquisition is conducted within 15 minutes after the start of these weather forecast periods. Missing data at these specified

time points are supplemented by the ISCCP (Young et al., 2018) using the best available image temporally closest to the target time period.

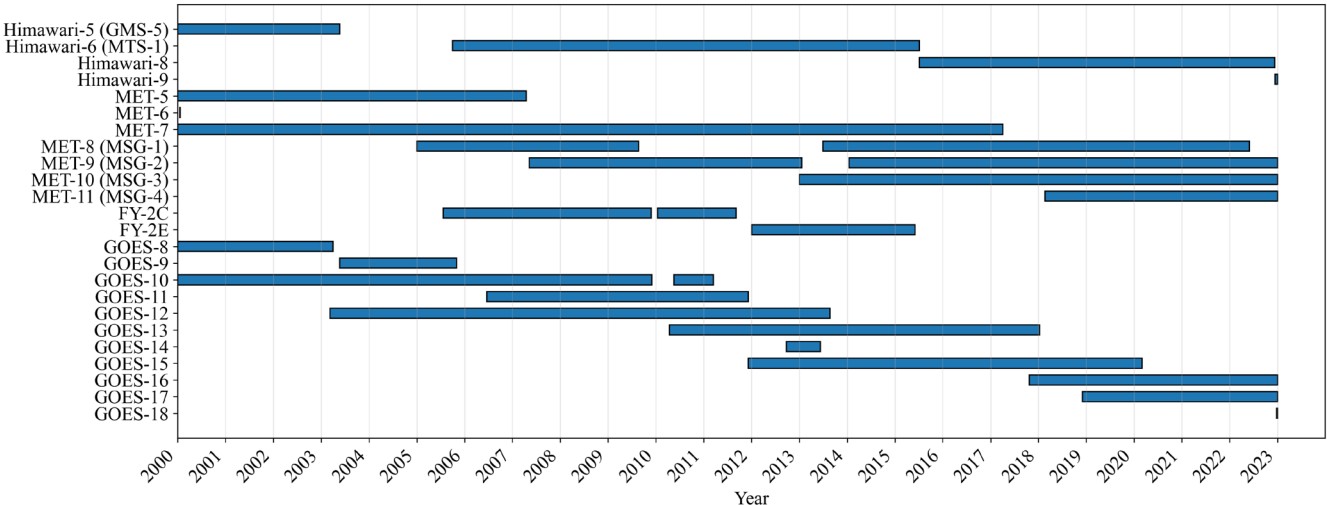

**Figure 1.** Gantt chart of 24 satellites within the research time interval and corresponding participation times: Himawari-5 (GMS-5),
Himawari-6 (MTS-1), Himawari-8, Himawari-9, MET-5, MET-6, MET-7, MET-8 (MSG-1), MET-9 (MSG-2), MET-10 (MSG-3), MET-11 (MSG-4), FY-2C, FY-2E, GOES-8, GOES-9, GOES-10, GOES-11, GOES-12, GOES-13, GOES-14, GOES-15, GOES-16, GOES-17, and GOES-18.

Due to the long time span and the large number of satellites and observation instruments involved, the primary
challenge in constructing global TOA infrared brightness temperatures lies in ensuring consistency. GridSat-B1 effectively addresses radiation calibration and temporal consistency issues through the following methods: (1) This dataset first uses ISCCP calibration methods to preliminarily integrate data from different satellites. Specifically for infrared (IR) channels, GridSat-B1 also implements a secondary calibration process. Secondary calibration utilises high-resolution infrared radiation

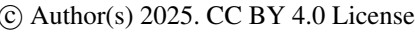

detectors (HIRS) as a reference standard, focusing on correcting systematic biases under low-temperature conditions. (2) To ensure the consistency and uniformity of long-term time series data, GridSat-B1 undergoes time normalisation processing. This process also utilises HIRS data as a calibration anchor point, effectively eliminating time offsets between different satellite observation systems, thereby significantly enhancing the temporal consistency of the entire IR brightness temperature historical dataset.

### 2.1.2. Polar orbit satellite data

MODIS comprises the Terra satellite, launched into a polar orbit in December 1999, and the Aqua satellite, launched into a polar orbit in April 2002. These satellites continuously collect data every 1–2 days across 36 spectral channels that cover the entire globe. Its exceptionally wide spectral range enables MODIS data to be used in a wide range of studies, including vegetation health, land cover, sea surface temperature, and cloud analysis (Hosen et al., 2023; Cai et al., 2011; Menzel et al., 2008). MOD represents Terra products, and MYD represents Aqua products. In this study, CLP, CTH, CER, and COT of MOD/MYD were used as training labels. Due to MODIS's lower orbital altitude and higher data quality, it is widely used as a ground truth label (Zhang et al., 2017).

The Cloud Profiling Radar and Cloud-Aerosol Lidar with Orthogonal Polarisation (CALIOP) lidar instrument, mounted on the Cloud-Aerosol Lidar and Infrared Pathfinder Satellite Observation (CALIPSO) satellite, was launched in April 2006 and ceased operations in June 2023. CALIPSO provides global vertical structure and characteristic observation data for aerosols and thin clouds (Zhang et al., 2017; Hagihara et al., 2010). The assessment of model accuracy primarily utilised cloud products from MODIS and CALIOP.

### 2.1.3. Meteorological field and auxiliary data

Considering the physical generation mechanism and development process of clouds, some meteorological fields and satellite IDs (Satid) constituting infrared brightness temperature data, as well as the corresponding satellite zenith angle (SZA), have been added to the input data. The parsing process for Satid and SZA in the auxiliary data is mentioned in the documentation for GridSat-B1 (Knapp et al., 2011). Additionally, the official documentation notes that the zenith angle correction for infrared brightness temperature images can be referenced in the work of Joyce et al. (2001).

Since the data from IRWVP and IRWIN grid fusion may originate from different satellites, Satid is divided into IRWVP satellite IDs (Satid_VP) and IRWIN satellite IDs (Satid_IN), and SZA is divided into IRWVP SZA (SZA_VP) and IRWIN SZA (SZA_IN). Meteorological field data are sourced from the European Centre for Medium-Range Weather Forecasts (ECMWF) Reanalysis v5 (ERA5) (Hersbach et al., 2020). This study selected ERA5 hourly air temperature profiles (ATP), relative humidity profiles (RHP), surface skin temperature (SKT), total column water vapour (TCWV), and soil type with a spatial resolution of 0.25°. To prevent unnecessary model redundancy caused by overly dense pressure levels



in the input data, which could affect the efficiency of model training and data production. ATP and RHP are each selected at four identical pressure levels: 1000, 850, 500, and 300 hPa.

### 2.1.4. Detailed information

Table 2 provides all the training and evaluation datasets used in this study. Due to the different institutional sources of the datasets, they may have different projection methods and spatio-temporal resolutions. To ensure the correct correspondence of pixels and data consistency, the data were first aligned to a unified 0.07° latitude and longitude grid before model construction. The nearest neighbour interpolation method (Huang et al., 2012) was used for resampling the MODIS Level-2 cloud product (MOD06/MYD06), while the bilinear interpolation method (Kim et al., 2019) was used for

resampling the ERA5 meteorological field. Due to differences in satellites and onboard sensors, there are differences in the spatial observation range and temporal observation frequency of the data. In this study, the input and target/evaluation data were matched in space and time to construct the infrared brightness temperature to cloud physical properties (IRBT2CPP) required for training. The following is a more detailed introduction to the data from different sources.

**Table 2.** Input, target and evaluation data preparation for building the dataset.

| | Variable | Source | Spatial resolution | Temporal resolution |
|---|---|---|---|---|
| Input | TOA Brightness Temperature (2 bands: IRWVP 6.7 and IRWIN 11 μm) | Gridsat | 8 km | 3 h |
| | Satellite zenith angle (SZA_VP and SZA_IN) | | | |
| | Satellite Index (Satid_VP and Satid_IN) | | | |
| | Surface skin temperature | ERA5 | 0.25° | 1 h |
| | Total column water vapor | | | |
| | Soil type | | | |
| | Air temperature profile (4 pressure levels: 1000, 850, 500, and 300 hPa) | | | |
| | Relative humidity profile (4 pressure levels: 1000, 850, 500, and 300 hPa) | | | |
| Target | Cloud phase | Aqua and Terra/MODIS | 1 km | 5 min |
| | Cloud top height | | | |
| | Cloud optical thickness | | | |



| Evaluation | Cloud effective radius | | | |
| | Cloud phase | Aqua and Terra/MODIS | 1 km | 5 min |
| | Cloud top height | | | |
| | Cloud optical thickness | | | |
| | Cloud effective radius | | | |
| | Cloud phase | CALIPSO/CALIOP | 1 km | - |
| | Cloud top height | | | |
| | Cloud optical thickness | | | |

## 2.2 Method

### 2.2.1. The main framework

The research technical route for AGCPP production are shown in Fig. 2. First, we match the infrared brightness temperature

of the GridSat-B1 and ERA5 meteorological field with the high-resolution cloud physical product of the MODIS satellite at the pixel level to construct a large-scale training sample IRBT2CPP. Then, we train the deep neural network cloud-based Attention-enhanced (At) UNet model (AtUNet) (Trebing et al., 2021) (CloudAtUNet) to learn the mapping relationship between brightness temperature and cloud physical parameters. Finally, we directly apply the model weights obtained from training to the entire GridSat-B1 brightness temperature time series to produce the AGCPP dataset, thereby achieving

continuous cloud physical parameter generation on a global scale and over a long time series. This method combines the advantages of polar-orbiting and geostationary satellites in terms of spatial resolution and observational continuity, while effectively overcoming inconsistencies caused by calibration, navigation, and sensor differences in multi-source data fusion.

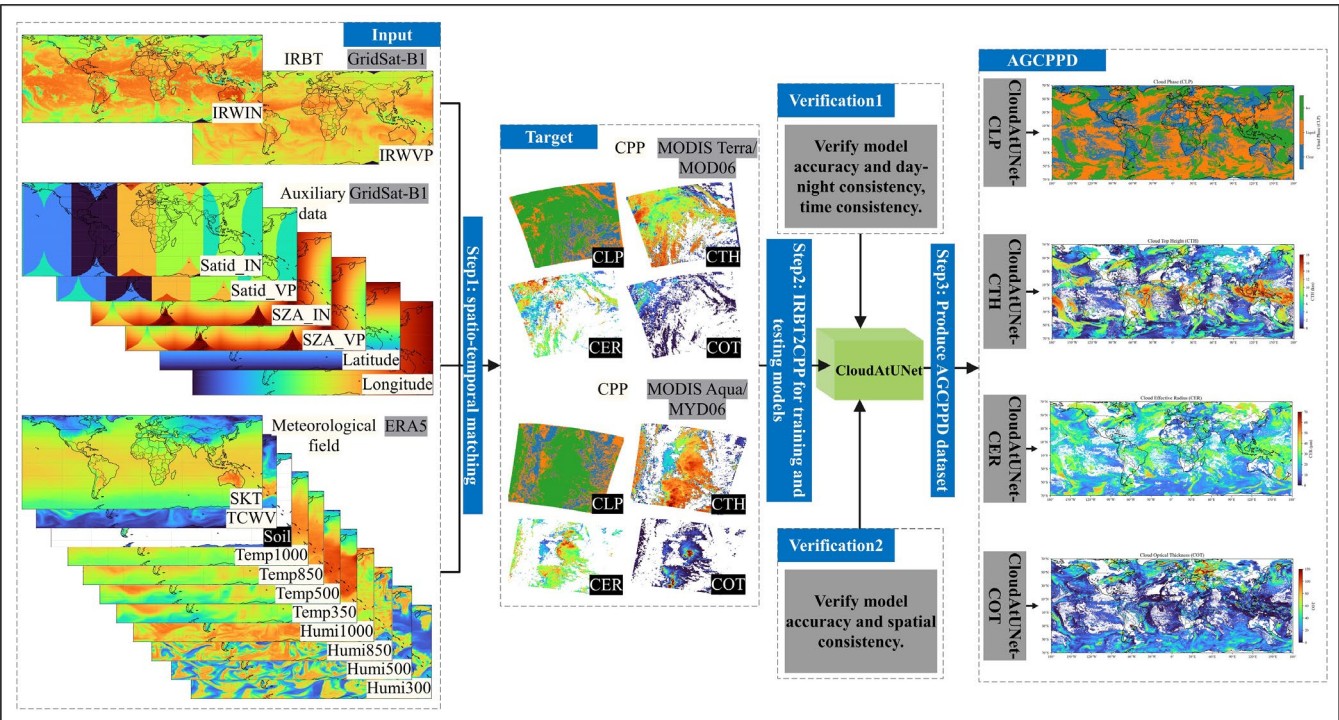

**Figure 2.** Flowchart of AGCPP production. It is worth noting that the input data consists of 19 channels, including infrared brightness temperature (IRBT), auxiliary data, and meteorological fields, while the target is CLP, CTH, COT, and CER images processed to the same resolution. The training input and output image sizes are both 64×64. For each target, this study conducted model training and validation and evaluation separately. Finally, a sliding window fusion strategy was used to produce global-scale cloud product data.

IRBT2CPP is a corresponding matching dataset of infrared brightness temperature and cloud products containing information from 2000 to 2022, spanning 23 years, with a total of approximately 700,000 samples. This is a fairly large dataset. Considering the lengthy and extensive data processing involved in constructing IRBT2CPP, we have also chosen to make this dataset publicly available(https://doi.org/10.57760/sciencedb.27171) (Zhao, 2025).

When training model parameters, data from 2000 to 2021 was used as the training-set, and data from the entire year of 2022 was used as the testing-set. These samples were evenly distributed in spatial dimensions, and the strategy for selecting the training and testing sets met the basic requirements of AtUNet (Trebing et al., 2021). This strategy and the large total sample size reduce the risk of overfitting together. In order to expand the training data, we applied data augmentation operations such as horizontal flipping, vertical flipping, and 90°, 180°, and 270° rotation to the training set, expanding the training-set to six times its original size. At the same time, the testing-set remained unchanged without any augmentation in order to objectively verify the effectiveness of data augmentation in improving model performance. In fact, data augmentation operations did improve accuracy. Specific information about the data is shown in Table 3.

**Table 3.** IRBT2CPP data amount statistics and training-set testing-set data amount division.



| | Total number of samples | Training-set | Testing-set | Training-set (data augmentation) | Testing-set (data augmentation) |
|---|---|---|---|---|---|
| GridSat2MOD | 373269 | 357744 | 15525 | 2146464 | 15525 |
| GridSat2MYD | 324950 | 312090 | 12860 | 1872540 | 12860 |
| MOD+MYD | 698219 | 669834 | 28385 | 4019004 | 28385 |

## 2.2.2. The machine learning model

Deep learning has achieved significant breakthroughs in the field of satellite remote sensing, particularly the UNet model, which is widely used in remote sensing image processing due to its exceptional spatial feature extraction capabilities (Liu et al., 2024a; Zhong et al., 2024). This study introduces an improved version of the UNet model called AtUNet (Trebing et al., 2021). This model incorporates the Convolutional Block Attention Module (CBAM) in the encoder section, reducing the number of parameters by 25% while maintaining the original UNet's accuracy, thereby significantly improving computational efficiency. Given that this model is specifically optimised for cloud physical retrieval, its input end integrates key physical prior parameters related to cloud formation (such as ERA5 temperature/humidity fields, etc.), we call it CloudAtUNet in this paper to highlight its embedded learning capability for cloud physics processes.

Figure 3 shows the complete structure of the CloudAtUNet model. Its encoder-decoder architecture synchronously captures the spectral response and spatial structural features of cloud systems through skip connections, making it particularly suitable for inverting cloud physical parameters with strong spatio-temporal inhomogeneity. The encoder learns the optimal nonlinear combination of multi-spectral brightness temperature and meteorological fields through convolutional layers, analysing sub-pixel-scale cloud physical attribute features while extracting the spatial distribution patterns of cloud systems. The attention mechanism CBAM module dynamically focuses on the core regions of cloud clusters while suppressing irrelevant meteorological background noise. The upsampling convolutional layers in the decoder preserve the fine-grained structural features of cloud boundaries, preventing spatial information loss during decoding and enabling precise spatial reconstruction of cloud parameters.

The model was trained using a local high-performance computing cluster and NVIDIA GeForce RTX 3090 graphics cards, applying CloudAtUNet to terabyte-scale GridSat-B1 brightness temperature data and Aqua/Terra MODIS cloud physical products. Key parameters for model training include: batch size = 512, maximum epochs = 300, and learning rate = 0.001. An early stopping strategy was used, stopping when the loss on the testing-set did not decrease by more than 0.1 for 15 consecutive epochs. All models stopped before reaching the maximum 300 epochs. The loss functions for model training differed. CrossEntropyLoss was selected for the CLP classification task, while MSELoss was selected for the CTH, COT, and CER regression tasks.



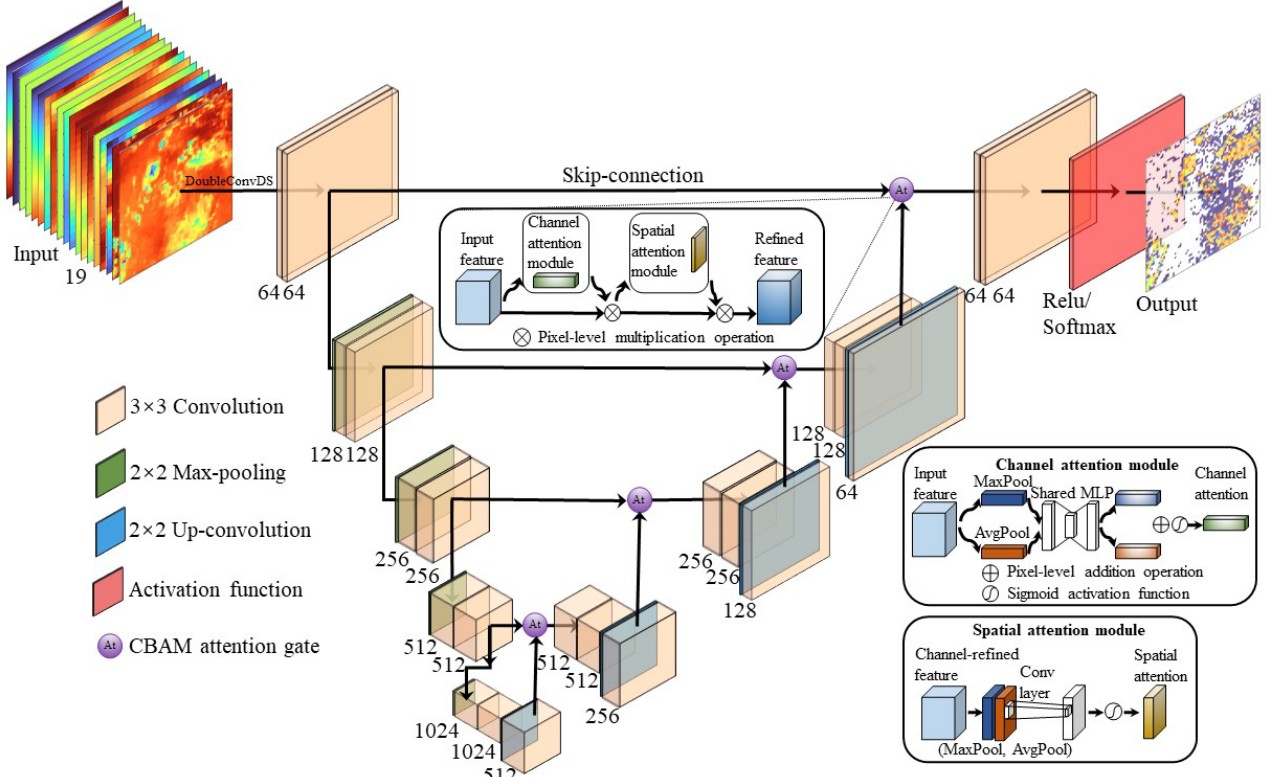

**Figure 3.** Image of the CloudAtUNet model structure, including the CBAM attention enhancement mechanism.

### 2.2.3. Image sliding window fusion strategy

Due to the large size of the images in the overlapping region between Gridsat-B1 and Aqua/Terra, they cannot be directly used as model inputs. Therefore, we divided each image into multiple small sample images with a matrix size of 64 × 64. In the actual production of AGCPP, each image takes approximately 2 minutes to process, so the 23-year product with 4 variables requires nearly 9,000 CPU hours (4 × 8 × 2 × 365 × 23 ÷ 60 = 8,955 h). Six RTX 3090 and four RTX A6000 GPUs were used to produce the dataset, taking nearly two months to complete.

The CloudAtUNet model was trained using a large number of image samples with a matrix size of 64×64 pixels. Therefore, when reconstructing the local prediction results of 64×64 pixels into a complete global cloud physical properties product image based on implicit prior knowledge (training weights), a certain stitching strategy needs to be adopted. To avoid gaps between adjacent prediction blocks, a sliding window fusion strategy based on linear weights is used.

First, the full image size: $H \times W$. PATCH: $P = 64$. STRIDE: $S = 10$. Since the image size may not be a multiple of

PATCH, it is necessary to use rounding up to fill in $H_2 \times W_2$:

$$H_2 = \left\lceil \frac{H}{P} \right\rceil P, W_2 = \left\lceil \frac{W}{P} \right\rceil P \qquad (1)$$



The starting coordinates and relative coordinates of the window are $(i, j)$ and $(x', y')$, respectively.

$$(i, j), \ 0 \le i \le H_2 - P,; 0 \le j \le W_2 - P \tag{2}$$

$$(x', y'), \ 0 \le x', y' \le P - 1 \tag{3}$$

275         The weight matrix is composed of piecewise linear functions in the row and column directions. When adjacent blocks exist at the window boundaries, the weights in the boundary regions transition linearly from 0 to 1 (left/top boundary) or from 1 to 0 (right/bottom boundary). By summing the weighted prediction values of the overlapping regions and normalizing them, a seamless stitched image is ultimately formed. The vertical weights $w_y^{(i)}(x')$ and horizontal weights $w_x^{(j)}(y')$ are respectively:

$$w_y^{(i)}(x') = \begin{cases} 0, & i = 0, \\ \frac{x'}{S-1}, & i > 0, 0 \le x' < S, \\ 1, & S \le x' < P - S, \\ \frac{P-1-x'}{S-1}, & i + P < H_2, P - S \le x' < P, \\ 0, & \text{Otherwise.} \end{cases} \tag{4}$$

$$w_x^{(j)}(y') = \begin{cases} 0, & j = 0, \\ \frac{y'}{S-1}, & j > 0, 0 \le y' < S, \\ 1, & S \le y' < P - S, \\ \frac{P-1-y'}{S-1}, & j + P < W_2, P - S \le y' < P, \\ 0, & \text{Otherwise.} \end{cases} \tag{5}$$

The final weight matrix $W$ is constructed by the outer product of vertical weights and horizontal weights:

$$W_{i,j}(x', y') = w_y^{(i)}(x') \cdot w_x^{(j)}(y') \tag{6}$$

Where $i$ and $j$ are the starting position indices of the window in the filled large image, $x' = x - i$, $y' = y - j$, $x', y' \in$

$[0, P - 1]$. The contributions of each window, $P_{i,j}$ (i.e., the predicted value at the relative coordinates $(x', y')$ within window $(i, j)$), are weighted and normalized to obtain the final image:

$$I(x, y) = \frac{\sum_{i,j} W_{i,j}(x-i, y-j) \cdot P_{i,j}(x-i, y-j)}{\sum_{i,j} W_{i,j}(x-i, y-j)} \tag{7}$$

## 3 Evaluation of the AGCPP

290         We conducted a systematic evaluation of the generated AGCPP, which was divided into temporal consistency, spatial consistency evaluations and diurnal consistency. In the temporal consistency assessment, the model was first evaluated using the official MODIS products for the entire year of 2022 as a benchmark to assess CLP, CTH, COT, and CER. Additionally, we used Aqua/Terra MODIS data from 2000 to 2022 to assess the annual accuracy of CLP, CTH, COT, and CER. Spatial



consistency assessment is also based on the official MODIS products for all years from 2000 to 2022, with errors statistically

calculated by longitude and latitude.

Then, we performed a diurnal consistency analysis using the official CALIOP product for the year 2022, focusing on evaluating the accuracy of the CLP, COT, and CER during the day and night. Lastly, we again performed a yearly temporal consistency evaluation using CALIOP to evaluate the annual accuracy performance of CLP, COT, and CER in the AGCPP with the official CALIOP products from 2006 to 2022. Since AGCPP uses Aqua/Terra MODIS as the training target, and the

official cloud product retrieval algorithms for MODIS and CALIOP are not identical, this may result in some systematic biases. Therefore, we also evaluated the annual accuracy performance of MODIS and CALIOP between the official cloud products CLP, COT, and CER from 2006 to 2022. To ensure the accuracy of the evaluation, the time difference between CALIOP and AGCPP was limited to ±2 minutes. The time difference between CALIOP and MODIS was also limited to ±2 minutes.


### 3.1. Time consistency evaluation with MODIS

In the evaluation, statistical error metrics for classification evaluation include Accuracy, Recall, Precision, and F1-score. Accuracy measures the proportion of correctly predicted samples out of the total number of samples, making it the most intuitive overall performance metric. Recall measures the proportion of actual positive examples correctly identified by the

model out of all true positive examples. Precision measures the proportion of samples predicted as positive that are actually positive. F1-score is the harmonic mean of Precision and Recall, serving as a balanced comprehensive evaluation metric between Precision and Recall, better reflecting the model's robustness. For regression error metrics, RMSE, MAE, MBE, $R^2$, and PearsonR are used. RMSE imposes heavier penalties on larger errors, reflecting the overall dispersion of prediction errors. MAE assigns equal weight to all errors, providing an intuitive measure of error magnitude. MBE is the difference

between the model's calculated result and the true value, helping to diagnose whether the model has systematic bias. $R^2$ measures the proportion of variance in the target variable that the model can explain, reflecting the model's goodness of fit. PearsonR measures the strength and direction of the linear relationship between predicted and true values.

Based on thresholds established in earlier publications for comparing the FY4A (AGRI) official cloud product, the Himawari-8 (AHI) official cloud product, and the TL-ResUnet–retrieved cloud product against MODIS, we applied the

thresholds to evaluate our model results (Zhao et al., 2024). In detail: For the evaluation of CLP for the cloud classification task, the Accuracy is 71.77% and 79.82% for FY4A and TL-ResUnet, respectively. For the CTH evaluation of the cloud regression task, the RMSEs of FY4A and TL-ResUnet are 3.58 and 1.99 km. For the COT evaluation, the RMSEs of Himawari-8 and TL-ResUnet are 14.62 and 12.87. For the CER evaluation, the RMSEs of Himawari-8 and TL-ResUnet are 10.14 and 10.14 μm respectively. Moreover, although PearsonR ranges from –1 to 1 with higher values indicating better

agreement, the quality limitations of the observational data mean that the threshold values for this metric differ among the





three retrieved cloud physical properties:for TL-ResUnet they are 0.884, 0.596, and 0.765, respectively (Tong et al., 2023; Li et al., 2023; Zhao et al., 2024).

### 3.1.1. CloudAtUNet model performance testing

CloudAtUNet-CLP uses 0, 1, and 2 to represent clear skies, water clouds, and ice clouds, respectively. Table 4 shows the statistical results of the error evaluation indicators on the testing-set (the whole year of 2022), and Fig. 4 shows the detailed data distribution under the testing-set evaluation. The evaluation results show that the Accuracy, Recall, Precision, and F1-score of the CLP classification are 0.823, 0.827, 0.827, and 0.827, respectively. Compared with the threshold indicators, it is higher than FY4A and TL-ResUnet's 71.77% and 79.82%. The closeness of the four metrics indicates that the model

achieves a good balance across the four metrics, demonstrating robust classification performance for both positive and negative category samples. Fig. 4 (a) shows the detailed data distribution of the CLP evaluation. It can be seen that the accuracy of clear sky forecasts is 81.52, water clouds are slightly lower at 80.21, and ice clouds are the highest at 86.38, which demonstrates that CloudAtUNet has good ability to analyse cloud physical properties in cloud classification tasks.

    Additionally, the RMSE, MAE, MBE, R², and PearsonR values for the CTH regression task are 1.617, 0.954, -0.039,

0.857, and 0.926, respectively. For the threshold metrics, RMSE is lower than FY4A and TL-ResUnet's 3.58 and 1.99, and PearsonR is higher than 0.884. MBE = -0.039 indicates that although CloudAtUNet-CTH slightly underestimates CTH (<0), the underestimation is minimal, and the model does not exhibit significant systematic bias. Considering the 0–18 numerical range of CTH in the official MODIS cloud product, this performance is highly commendable. Fig. 4(b) shows the scatter density distribution diagram for CTH assessment. It can be seen that the majority of data are concentrated within 0–4 km,

with a significant portion also clustered around the 1:1 line.

    Secondly, the RMSE, MAE, MBE, R², and PearsonR values for the COT regression task are 11.314, 6.871, -0.649, 0.381, and 0.727, respectively. For threshold indicators, RMSE is lower than Himawari-8 and TL-ResUnet's 14.62 and 12.87, and the PearsonR is well above 0.596. MBE = -0.649 indicates that while CloudAtUNet-COT underestimates COT, considering the 0–150 numerical range in the official MODIS cloud product, the degree of underestimation is acceptable and

does not indicate severe systematic bias. Fig. 4(c) shows the scatter density distribution diagram for COT evaluation, with the majority of data concentrated between 0 and 20. For data above 20, the scatter density distribution diagram for CloudAtUNet-COT becomes more dispersed, which is the primary source of error.

    Finally, the RMSE, MAE, MBE, R², and PearsonR values for the CER task are 7.181, 5.133, -0.132, 0.638, and 0.799, respectively. For threshold indicators, RMSE is lower than Himawari-8 and TL-ResUnet's 10.14 and 10.14 μm, and

PearsonR is higher than 0.765. The MBE of -0.132 still indicates that CloudAtUNet-CER slightly underestimates CER, with the underestimation being greater than CTH but less than COT. This is primarily due to the 0–60 numerical range of CER in the official MODIS cloud product, so there is no significant systematic bias. Fig. 4(d) shows the scatter density distribution diagram for CER evaluation, indicating that CER values are primarily concentrated in the 7–20 μm and 30–45 μm ranges.



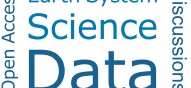

The former primarily targets the retrieval of water clouds, while the latter primarily targets the retrieval of ice clouds (Liu et al., 2023). CloudAtUNet-CER demonstrates slightly stronger retrieval capability for water cloud CER than for ice cloud CER.

**Table 4.** Error indicators for the evaluation results of the testing-set.

| Productions | Indicators | | | |
| | Accuracy | Recall | Precision | F1-score |
| --- | --- | --- | --- | --- |
| CLP | 0.823 | 0.827 | 0.827 | 0.827 |

| | Indicators | | | | |
| | RMSE | MAE | MBE | $R^2$ | PearsonR |
| --- | --- | --- | --- | --- | --- |
| CTH | 1.617 | 0.954 | -0.039 | 0.857 | 0.926 |
| COT | 11.314 | 6.871 | -0.649 | 0.381 | 0.727 |
| CER | 7.181 | 5.133 | -0.132 | 0.638 | 0.799 |





**Figure 4.** Detailed distribution of testing-set evaluations. (a) Confusion matrix for CLP. Scatter density distribution diagram under the kernel density estimation of (b) CTH. (c) COT. (d) CER.

### 3.1.2. CloudAtUNet annual evaluation

This section uses Aqua/Terra MODIS data from 2000 to 2022 to evaluate the annual accuracy of CLP, CTH, COT, and CER. We expect the results of the CloudAtUNet model to remain consistent in annual assessments, but in reality, the accuracy metrics for each year cannot be completely consistent. This section evaluates accuracy using annual MODIS official cloud



product data, providing users with some reference for using AGCPP, especially when using data from a specific year. Fig. 5 shows the evaluation results for CLP, and Fig. 6 shows the evaluation results for CTH, COT, and CER.

It can be seen that the accuracy index of all years in the training-set in Fig. 5 is approximately 0.85. Although there are slight fluctuations in accuracy from year to year, with slight differences, such as 2004 being slightly lower and 2010 being slightly higher. When users need to use AGCPP, they can use this chart to roughly determine whether the accuracy of the data usage year is slightly higher or slightly lower than the surrounding years. Additionally, the fluctuations in all data in the training-set are almost within three decimal places, so it can be said that the CloudAtUNet-CLP model has good learning and

cloud classification capabilities, and the results of the annual tests are relatively continuous and consistent, with no extreme anomalies. The results on the testing-set are slightly lower than those on the training set, which is normal because the model has not seen the samples in the testing-set. A detailed analysis has been presented in 3.1.1 and will not be described here.

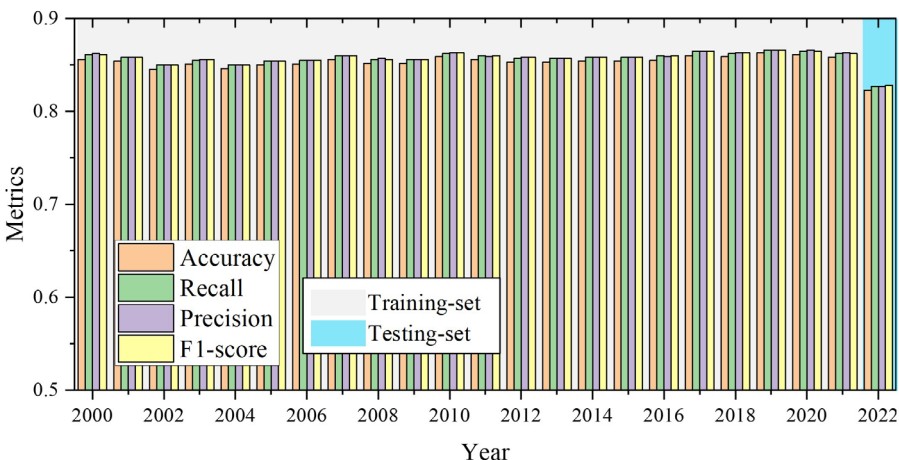

**Figure 5.** Evaluation of CLP classification accuracy on training-set and testing-set based on annual Aqua and Terra MODIS official cloud products.

     Figure 6 shows the statistical results of the regression metrics for CTH, COT, and CER. Since RMSE and MAE indicate higher model accuracy when their values are lower, they are plotted on the same graph. $R^2$ and PearsonR also indicate higher

model accuracy when their values are higher, so they are also plotted on the same graph. Finally, MBE is plotted on a separate graph. In Fig. 6 (a), the RMSE of the CTH training-set is around 1.5, while the testing-set is slightly higher at around 1.6. The same is true for the RMSE in Fig. 6 (b) and Fig. 6 (c). Through the MBE indicator (model result minus true value), we also found that whether it is the training-set or the testing-set, the model tends to give smaller values in regression prediction, slightly underestimating. Finally, although MSE was used as the loss function during training, $R^2$ and PearsonR

were more stable than RMSE and MAE in annual statistics. For example, the $R^2$ of all years in the CTH training-set was almost within the range of 0.88±0.01, while the annual RMSE and MAE varied to a certain extent. However, all statistical results and data prove that the CloudAtUNet annual detection results are stable.

**Figure 6.** Evaluation of (a) CTH. (b) COT. (c) CER. regression results based on training-set and testing-set of official Aqua and Terra MODIS cloud products.

## 3.2. Spatial consistency evaluation with MODIS

This section presents the spatial distribution of spatial differences between AGCPP and MODIS for all years from 2000 to 2022. Fig. 7 shows the spatial distribution of spatial differences between AGCPP and MODIS along latitude, with statistical results averaged every 3°. Fig. 8 shows the spatial distribution of differences between AGCPP and MODIS along longitude, with statistical results averaged every 5°. Fig. 7 and Fig. 8 present the mean values (blue dots), one standard deviation (STD) above and below the mean (blue shading), and box plots (upper bound, lower bound, upper quartile, lower quartile, and median). Firstly, the latitudinal distribution of CLP is less different, but the classification effect is slightly lower at low



latitudes than at high latitudes in Fig. 7(a). This is mainly due to the complex cloud structure in the tropics (e.g., cumulonimbus, high convective clouds), which makes it easier to confuse water clouds with ice clouds. Although deep learning models well trained, may still misclassify when encountering inhomogeneous cloud phases in the tropics. This phenomenon is mentioned in the study of Meyer et al. (2016), which points out that especially in tropical strong convective clouds with strong ice-water mixing, leading to more significant classification errors.

Secondly, the spatial error distributions of CTH and CER in Fig. 7(b) and Fig. 7(d) are relatively consistent, with higher values near the equator than in other regions. This is primarily because the CTH and CER values near the equator are inherently larger, resulting in correspondingly larger error margins during model evaluation. Mitra et al. (2021) compared MODIS CTH with Lidar observations and found that for CTH, errors are smaller for low clouds and larger for high clouds. Since cloud heights near the equator are inherently higher, the higher cloud heights result in larger errors in CTH retrieval.

Zhang et al. (2025) also found that in the equatorial and low-latitude regions, due to high ice/liquid cloud mixing and complex infrared absorption patterns, CER values are higher, resulting in larger errors during model evaluation.

Finally, as shown in Fig. 7(c), the spatial error distribution of COT is smaller near the equator and increases gradually with increasing latitude. This is primarily because ice clouds are more abundant in mid-to-high latitude regions, and multiple studies have shown that ice clouds typically have higher optical thickness than liquid water clouds (Takahashi et al., 2016).

Additionally, Alexandrov et al. (2025) also pointed out that if the 3D radiative effects are ignored in cloud retrieval calculations, the COT of ice clouds at high latitudes would be significantly underestimated. Therefore, in these regions, both the MODIS official algorithm and our deep learning model face greater challenges in estimating COT.

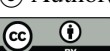

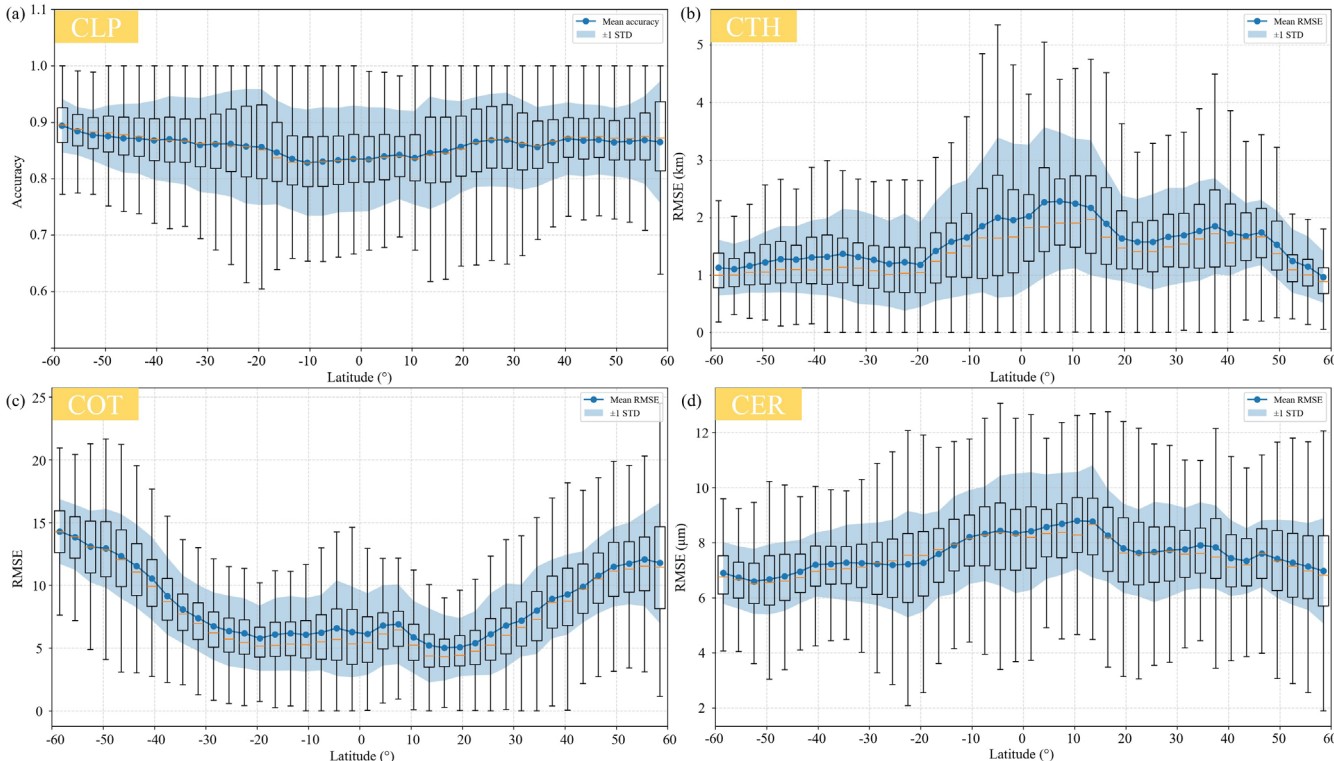

**Figure 7.** Distribution of spatial differences in latitude between AGCPP and MODIS. (a) CLP accuracy. (b) CTH RMSE. (c) COT RMSE. (d) CER RMSE.

Figure 8 shows that the STD of the longitude-direction metrics for all products exhibits a distinct peak-trough cycle, with eight peaks occurring at –135°, –90°, –45°, 0°, 45°, 90°, 135°, and 180°, and the midpoints between the peaks representing the trough values. This pattern is due to the fact that the AGCPP is derived from the GridSat-B1 product, which is constructed by merging the observed brightness temperatures from multiple geostationary satellites. At the meridian point (subsatellite point) of each satellite, the observational geometry and calibration are most consistent, resulting in the lowest errors and STDs. As the zenith angle of the same satellite increases, the STD also gradually increases. However, once the observation area crosses into the coverage zone of an adjacent satellite, discontinuities in radiometric calibration and geometric alignment occur, introducing additional errors and resulting in STD peaks at these longitude positions. Gunshor et al. (2009) demonstrated that inter-calibration errors between satellites primarily stem from differences in temporal alignment, spatial resolution, and geometric alignment, with these factors being most pronounced at the mosaic boundaries. Although GridSat-B1 has undergone calibration and navigation operations, it can only minimise differences in radiometric calibration and geometric alignment rather than completely eliminate them, resulting in this phenomenon.





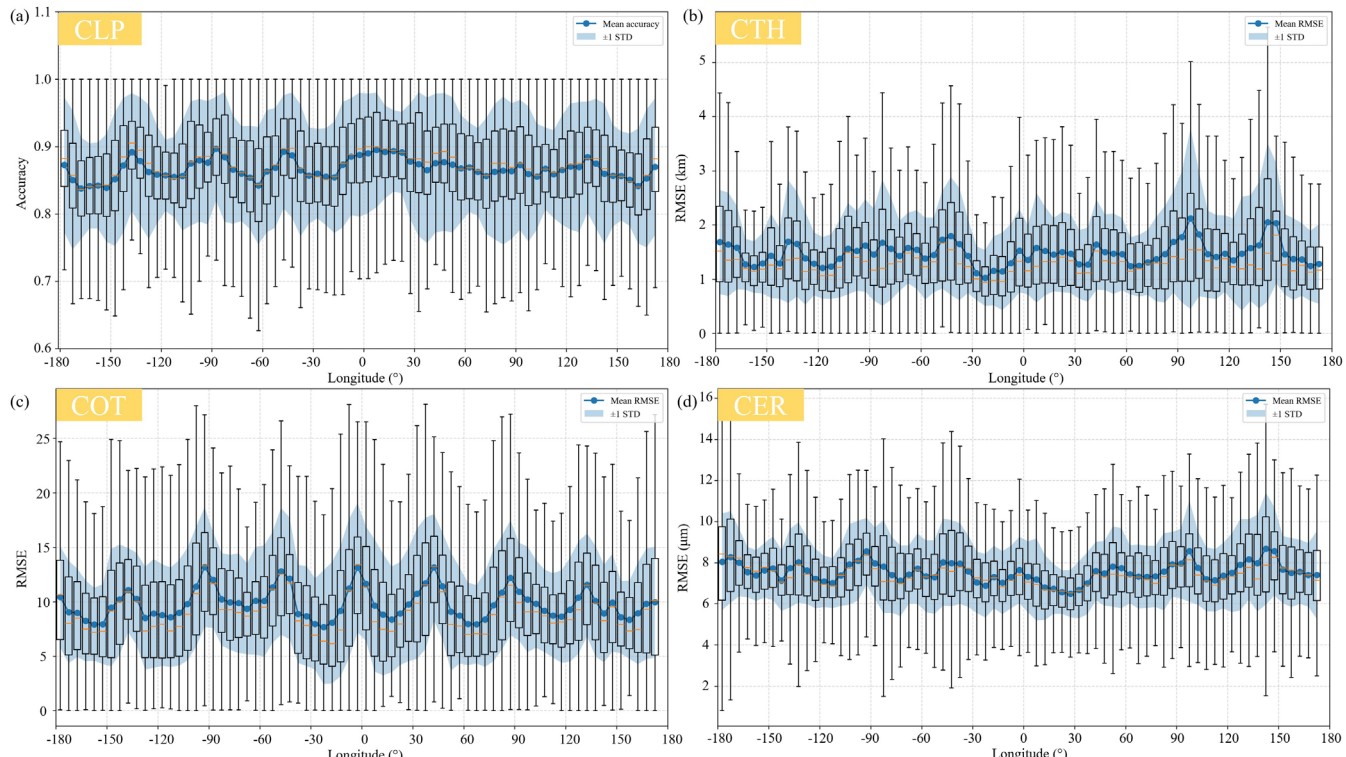

**Figure 8.** Distribution of spatial differences in longitude between AGCPP and MODIS. (a) CLP accuracy. (b) CTH RMSE. (c) COT RMSE. (d) CER RMSE.

## 3.3. Day-night and time consistency evaluation with CALIPSO/CALIOP

### 3.3.1. Evaluation of day-night consistency

In this study, we evaluated the consistency of daytime and nighttime accuracy of the CloudAtUNet model results using CALIPSO/CALIOP. Additionally, the comparison results between the FY4A (AGRI) official cloud product, the Himawari-8 (AHI) official cloud product, and the CALIOP official cloud product are provided in the literature (Li et al., 2023; Zhao et al., 2024). Specifically, on CLP, the accuracy value of AHI compared to CALIOP is 0.736 (Zhao et al., 2024); On CTH, the RMSE value of AGRI and CALIOP results is 4 (Zhao et al., 2024); On COT, the RMSE value of AHI and CALIOP results is 23.71 (Li et al., 2023).

Figure 9(a) shows that compared with the active sensors CALIPSO/CALIOP, the accuracy of the CloudAtUNet model CLP during the day and at night is 0.787 and 0.775, respectively. Compared with the official cloud product 0.736 of Himawari-8 (AHI), the accuracy of the CLP product output by the CloudAtUNet model is higher, and there is little difference in accuracy between day and night, indicating good consistency between day and night. Secondly, Fig. 9(b) shows that the daytime and nighttime accuracies of the CloudAtUNet model CTH are 3.384 and 3.568, respectively. Compared to



the official cloud product FY4A (AGRI) with an accuracy of 4, the CloudAtUNet model's CTH product has higher accuracy, and there is little difference in accuracy between daytime and nighttime, demonstrating good consistency between day and night. Finally, Fig. 9(c) shows that the daytime and nighttime accuracies of the CloudAtUNet model's COT product are 16.87 and 17.936, respectively. Similarly, using the Himawari-8 (AHI) official cloud product's accuracy of 23.71 as the standard, the CloudAtUNet model's COT product accuracy is higher. Overall, due to the use of infrared brightness temperature and the day-night unified CloudAtUNet model, the model results exhibit good day-night consistency.

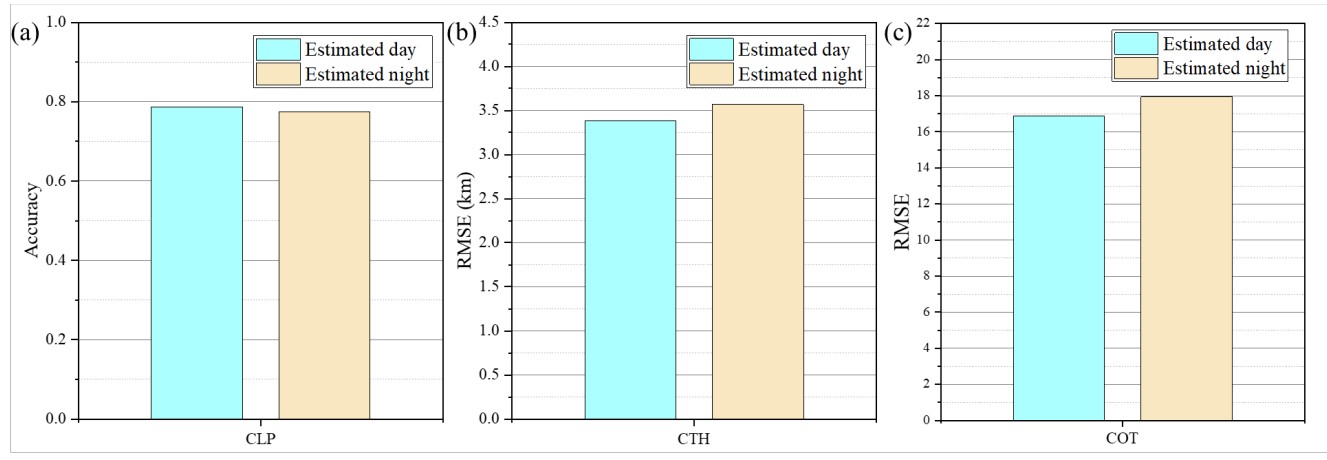

**Figure 9.** The accuracy of the CloudAtUNet model (daytime and nighttime) was evaluated using data from the active sensors CALIPSO/CALIOP. (a) CLP accuracy. (b) The RMSE of CTH. (c) The RMSE of COT. The blue bar chart shows the model's daytime accuracy, and the yellow bar chart shows the model's nighttime accuracy.

### 3.3.2. Annual accuracy evaluation between AGCPP and CALIPSO/CALIOP

After producing the AGCPP dataset for 2000–2022, we matched the AGCPP with CALIPSO/CALIOP data for the overlapping time period and conducted an evaluation. To ensure the accuracy of the evaluation, the time difference between CALIOP and AGCPP was limited to ±2 minutes. Additionally, since AGCPP is labelled using Aqua/Terra MODIS as the training target, and the official cloud product retrieval algorithms for MODIS and CALIOP are not identical, this may introduce some systematic biases. Therefore, we also conducted an evaluation for comparison, with the time difference between CALIOP and MODIS restricted to ±2 minutes.

Figure 10 shows the comparison results of AGCPP-CALIOP (G-C) and MODIS-CALIOP (M-C) for the three variables CLP, CTH, and COT. It can be seen that the accuracy of G-C is generally slightly lower than that of M-C in Fig. 10(a). The classification accuracy of M-C is approximately 0.8, while that of G-C is around 0.75. This is similar to the evaluation results in Wang et al. (2016) (the consistency between MODIS and CALIOP was 77.8%). In years where M-C has higher accuracy, G-C also has higher accuracy, as the labels used for AGCPP training are from MODIS rather than CALIOP.



Therefore, some of the errors in G-C's evaluation are due to errors between M-C. Fig. 11 shows the confusion matrix information of the detection results. It can be seen that the M0-C0 values for 2013 and 2015 are 68.20 and 65.64, respectively, which are slightly lower than those of other years. Similarly, the G0-C0 values for 2013 and 2015 are 64.79 and 64.65, respectively, which are also slightly lower than those of other years. This further indicates that the results of the G-C evaluation are to some extent dependent on the gap between M and C. After being trained using MODIS as the label, AGCPP effectively captures and fits the MODIS information, with consistent and continuous accuracy year by year.

Additionally, the accuracy of the regression tasks CTH and COT is compared using RMSE, as shown in Fig. 10(b) and Fig. 10(c), respectively. Similarly, the regression accuracy of G-C is slightly lower than that of M-C. Since a smaller RMSE value indicates higher model accuracy, the value for G-C is slightly larger. However, the annual accuracy is relatively consistent and continuous for both CTH and COT. References also note that the errors in CTH for MODIS and CALIOP are significant, for example, the bias for pixels larger than 2 km exceeds 3 km (Weisz et al., 2007), which aligns with our statistical results.



500

**Figure 10.** Annual evaluation results of AGCPP and CALIOP (G-C) and annual evaluation results of MODIS and CALIOP (M-C). M stands for MODIS, C stands for CALIOP, and G stands for AGCPP. (a) Comparison of CLP accuracy. (b) Comparison of CTH RMSE. (c) Comparison of COT RMSE.



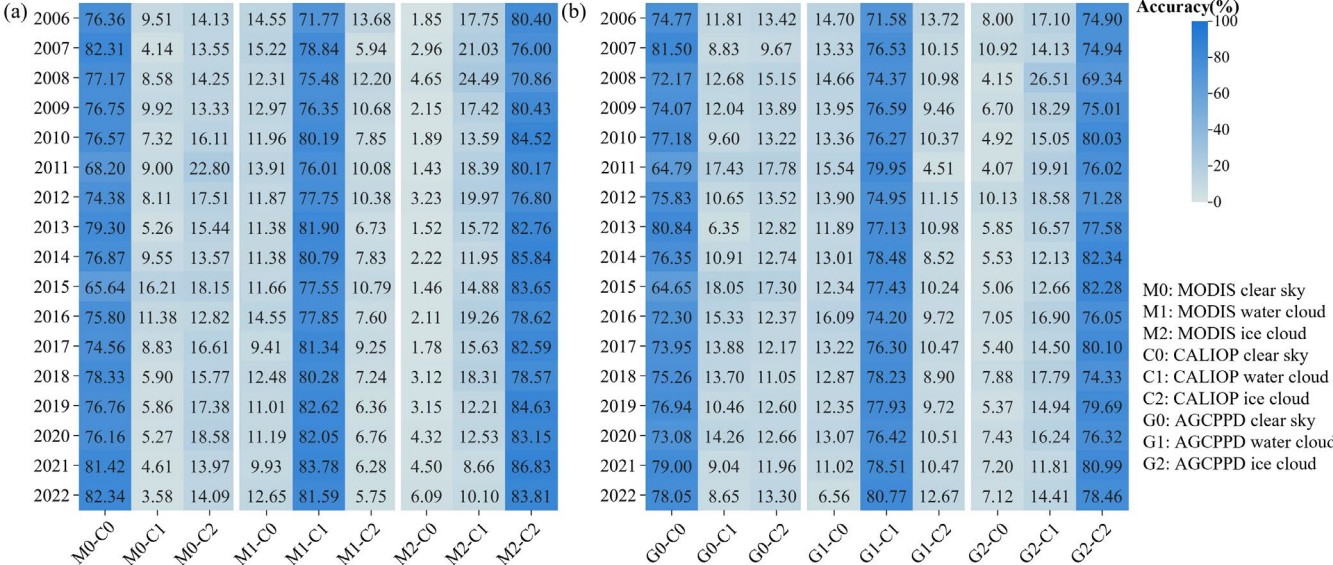

**Figure 11.** Confusion matrix for CLP model evaluation. M stands for MODIS, C stands for CALIOP, and G stands for AGCPP. 0, 1, and 2 represent the pixel being classified as clear sky, water cloud, and ice cloud in the model, respectively. (a) Evaluation results of MODIS official cloud products and CALIOP official cloud products. (b) Evaluation results of AGCPP and CALIOP official cloud products.

## 4 Basic characterization of AGCPP

AGCPP is also currently the world's first dataset capable of providing high spatio-temporal resolution cloud physical property products for the period 2000-2022. Since MODIS is located on polar-orbiting satellites, the spatial coverage of cloud products at the same time is insufficient, making it difficult to conduct large-scale synchronous observations of the entire globe. Therefore, the spatio-temporal distribution characteristics of the obtained cloud physical properties lack representativeness.

Here, we initially defined and calculated three physical quantities characterising cloud cover based on the CLP product: ice cloud fraction (ICF), water cloud fraction (WCF), and total cloud fraction (TCF) (Zhao et al., 2024). Since MODIS is insufficient to cover the entire globe at a 3 h time resolution, especially in the equatorial regions. Due to its 3 h temporal resolution, MODIS is insufficient to cover the entire globe, particularly in equatorial regions. Therefore, we employed a 1.5 h temporal resolution, averaging MODIS transit time images to obtain monthly average MODIS (MODIS transit time) images, as shown in the figure. Similarly, AGCPP also selected images with the same temporal and spatial coverage to obtain monthly average AGCPP (MODIS transit time) images. Missing time intervals were interpolated using adjacent time points. Finally, we also calculated the monthly average images of the AGCPP (All-day global) product.

By calculating the seasonal averages (DJF, MAM, JJA, SON) of the monthly average results for each year from 2020 to 2022, we plotted Fig. 12, Fig. S 1, Fig. S 2, Fig. S 3, and Fig. S 4 (see Supplementary Materials Figure S1–S4). Based on the



seasonal average results, we further calculated the annual average results for 2020–2022, as shown in Fig. 12. By comparing

the results, it can be seen that the MODIS official product and the AGCPP product are highly consistent in terms of ICF,

WCF, TCF, CTH, COT, and CER at the MODIS transit time, as shown in Fig. 12(a,d,g,j,m,p) and Fig. 12(b,e,h,k,n,q),

especially in the equatorial regions. However, since the Terra/MODIS overpass time is approximately 10:30 AM local time,

while the Aqua/MODIS overpass time is approximately 1:30 PM local time, the spatiotemporal distribution characteristics of

the obtained cloud physical properties lack representativeness. Nevertheless, the all-day global product of AGCPP, which

has global spatial coverage, can compensate for this deficiency. This is significantly different from the official MODIS

products, as shown in Fig. 12(a,d,g,j,m,p) and Fig. 12(c,f,i,l,o,r).



**Figure 12.** Annual average spatial distribution of global cloud physical products from 2020 to 2022, respectively: ICF (a–c), WCF (d–f), TCF (g–i), CTH (j–l), COT (m–o), and CER (p–r). Vertically, these are the seasonal averages of the MODIS official cloud products with a 1.5 h transit time interval (a, d, g, j, m, p), the seasonal averages of AGCPP at the same MODIS transit time and coverage location (b, e, h, k, n, q), and the seasonal averages of AGCPP's all -day global products at all times (c, f, i, l, o, r).





Therefore, based on the spatio-temporal continuity and high accuracy of AGCPP products, this paper can comprehensively and accurately analyse the physical characteristics of clouds on a global scale. Here, we have compiled statistics on the changes in cloud cover frequency (CCF), CTH, COT, and CER over a period of 23 years between latitudes 60°S and 60°N. The basic cloud feature analysis based on AGCPP is shown in Fig. 13, with statistics separated for the Northern Hemisphere and Southern Hemisphere. For the Northern Hemisphere, the seasons are spring (MAM), summer

(JJA), autumn (SON), and winter (DJF). For the Southern Hemisphere, the seasons are reversed: spring (SON), summer (DJF), autumn (MAM), and winter (JJA). As shown in Fig. 13(a)(e), CCF indicates that cloud cover in all seasons is higher in the Southern Hemisphere than in the Northern Hemisphere. Additionally, the CTH values for the seasons in the Northern Hemisphere, ranked from highest to lowest, are summer, autumn, spring, and winter, which is completely consistent with the seasonal distribution of CTH in the Southern Hemisphere, as shown in Fig. 13(b)(f). The COT values in the Northern

Hemisphere are ranked from highest to lowest as winter, autumn, spring/summer, while in the Southern Hemisphere they are winter, autumn/spring, summer, as shown in Fig. 13(c)(g). Finally, the CER values in the Northern Hemisphere are higher in autumn/winter than in spring/summer, while in the Southern Hemisphere they are consistent with the Northern Hemisphere, also higher in autumn/winter than in spring/summer, as shown in Fig. 13(d)(h).





**Figure 13.** Seasonal average change curves for CCF, CTH, COT, and CER for different years between latitudes 60°S-60°N from 2000 to 2023 based on AGCPP products. Statistics are divided into the northern and southern hemispheres. The black, red, blue, and green lines represent MAM, JJA, SON, and DJF, respectively. (a) and (e) are CCF variables. (b) and (f) are CTH variables. (c) and (g) are CTH variables. (d) and (h) are CTH variables.





## 5 Data availability

The AGCPP data described in this paper have been made publicly available. We would like to express our special thanks to Science Data Bank and its staff for their help and support in the data publication process. All data can be accessed at https://doi.org/10.57760/sciencedb.26292 (Zhao et al., 2025).


## 6 Conclusion

In order to provide a full-time, long time series, and complete continuum of global cloud physical property products, a dataset called AGCPP (containing CLP, CTH, COT, and CER products) was produced, covering latitudes from -70° to 70°, and longitudes from -180° to 180°. AGCPP provides data for every 3 h throughout the day from 2000 to 2022, with a spatial

resolution of 0.07° for cloud physical properties. In this study, we evaluated the AGCPP data quality with the official Aqua and Terra/MODIS (MOD06/MYD06) cloud product and the official CALIPSO/CALIOP 1km cloud product. The results showed that AGCPP passed the temporal consistency assessment, the diurnal consistency assessment and the spatial consistency assessment. In addition, MODIS official cloud products and AGCPP were compared in terms of seasonal and annual averages using the same MODIS transit time and coverage area to ensure consistency. Furthermore, AGCPP all-day

global spatio-temporal coverage was used to compensate for the lack of representativeness of MODIS spatial distribution. Finally, a simple statistical analysis of the physical characteristics of clouds in the northern and southern hemispheres was performed based on the AGCPP all-day global product. Due to the long time-series and all-day global nature of the dataset, it is expected that the dataset AGCPP will significantly increase the potential for climate change studies, especially on the potential feedback effects between clouds, surface albedo, and radiation. Considering the lengthy and extensive data

processing    involved    in    constructing    IRBT2CPP,    we    have    also    chosen    to    make    this    dataset    publicly available(https://doi.org/10.57760/sciencedb.27171) (Zhao, 2025). We look forward to researchers worldwide building upon our work to iterate and develop cloud physical property products with higher accuracy, thereby collectively advancing research in the field of cloud physical properties. In addition, more research needs to be conducted in the future regarding how to complement the data on cloud physical properties products at the North and South poles.


## Author contribution

Zhao, L.: Conceptualization, Methodology, Writing–original draft, Investigation, Validation, Data curation, Drawing. Zhang, F.: Supervision, Writing-review & editing, Funding acquisition. Li, J.: Investigation, Validation. Lu, F.: Supervision, Discussion. Zhao, Z.: Investigation, Discussion. All authors reviewed the manuscript.




**Competing interests**

The authors declare that they have no conflict of interest.

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
