# Peer review of "AGCPP: All-day Global Cloud Physical Properties dataset with 0.07° resolution retrieved from geostationary satellite imagers covering the period from 2000 to 2022"

_Earth System Science Data, 2025_

## Referee Comment (RC2)

Paper number: essd-2025-425

"AGCPP: All-day Global Cloud Physical Properties dataset with 0.07° resolution  retrieved from geostationary satellite imagers covering the period from 2000 to 2022" (Zhao et al.)

This paper describes a new satellite-based global cloud dataset consisting of cloud phase, top height, optical thickness, and effective radius, called "All-day Global Cloud Physical Properties (AGCPP)", which is designed to overcome limitations in existing cloud data products by providing continuous, global coverage of essential cloud properties from 2000 to 2022, with high spatial and temporal resolution. This was achieved by using a deep learning model, CloudAtUNet, which was trained on MODIS and applied to gridded geostationary satellite data (GridSat-B1). The work included evaluation results against MODIS and CALIOP products. The paper presents an ML-based approach to generate consistent global cloud products utilizing existing satellite data, which would potentially provide benefits for climate change research, particularly by enabling long-term, all-day analysis.

Additional details and results would be desirable to strengthen the physical interpretation of the model training, particularly given the use of only two IR channels. In particular, for nighttime evaluation, further discussion of cloud optical properties such as cloud optical thickness and effective radius would enhance the scientific robustness of the work for publication.

The authors present yearly trends of the AGCPP products in Figure 13. To better highlight the strengths of this dataset, it would be beneficial to include an additional comparison with existing global cloud datasets. While comparison with ISCCP or SatCORPS may not be feasible due to the lack of overlapping periods, a comparison with CLARA-A3 or current ABI/AHI products for specific regions could provide valuable context and validation.

Major comments

MODIS L2 cloud products have been used in the initial CloudAtUNet model training, which would be anticipated to produce degraded cloud optical properties at night due to lack of visible information. Although the authors designed AGCPP generating day and night cloud products utilizing both GridSat and ERA-5 and show consistent trends against MODIS and CALIOP (CLP and COT), it does not provide clear explanations about using two IR bands only and estimation of nighttime evaluation results, especially for comparisons with CALIOP data (such as cloud optical thickness and effective radius). More discussions

about how to match with MODIS and CALIOP products and physical explanations about nighttime comparisons (1-2 additional figures) would be needed.

Minor comments / typos

Section 2.2.1 and 2.2.2 have several overlaps, which could be better reorganized.

Section 5 Data availability could be placed after Section 6.

Line 57: "Wang et al. 2024 -> Better references such as Bessho et al. 2016

Line 84-86: Complete the sentence "For example, calibrating raw sensor signals to radiant ..."

Line 99: "in the fact that AGCPP is the first global dataset with a spatial resolution of 0.07° and a temporal resolution of 3 h."  -> The statement would be better revised with additional specifications regarding what distinguishes this dataset. It may be clarified whether the emphasis lies on its comparatively high spatial and temporal resolutions relative to existing global cloud datasets such as ISCCP, NASA's SatCORPS, and CLARA-A3, for instance, if the authors intended. Also, specifying that it is a "satellite-based" global cloud dataset would provide important context of the description.

Table 1: "specifications of" will be better than "Comparison results". Please spell out the acronyms of the datasets.

Line 111-112: Revise the sentence. Maybe evaluation would be "evaluate"?

Line 161: "MOD/MYD" -> I can guess those are from collection 6, but specify the data collection version with a proper reference here first.

Line 180-181: Correct the incomplete sentence.

Table 3: Correct the first row.

Line 260 and Table 2: Gridsat -> GridSat

Line 299-304: Please add more details on how the initial cloud detection has been treated, which are also different between MODIS and CALIOP.

Line 411: Missing words. Complete the sentence "Although deep learning models well trained...".

Line 516-517: Correct the sentence "Since MODIS is insufficient to cover the entire globe at a 3 h time resolution, especially in the equatorial regions..."

Figure 13: Correct the caption. CTH has been repeated.

Line 567: Add satellite-based or satellite in "global cloud physical ..."

---

## Author Comment (AC2)

**Response to RC2**

Response for the Editor:

We extend our gratitude to the reviewers and editors for their patient scrutiny. In our response, the reviewers' questions appear in black typeface, while our answers are uniformly presented in blue typeface. To facilitate comprehension, content quoted from the main text of the paper is rendered in blue italics. Finally, following each quotation, the relevant chapter and paragraph numbers are indicated in red typeface.

Regarding the article title, we have made certain modifications, changing it to "All-day global cloud physical properties products with 0.07° resolution retrieved from geostationary satellite imagers covering the period from 2000 to 2022." The product name used throughout the manuscript has also been amended to "DaYu-GCP." During the revision process, BG participated in discussions and manuscript editing, while WL calculated theoretical brightness temperatures using radiative transfer modelling, thereby providing the experimental data foundation for our physical interpretation analyses. Consequently, these two authors have been added to the original author list. Furthermore, ZZ contributed additional insights and devoted substantially more time to this revision. Therefore, in accordance with the paper's contribution protocol, the author order of ZZ and JL has been adjusted. All authors have no reservations regarding the final author sequence. Beyond addressing the two reviewers' comments, all authors engaged in thorough discussions and meticulous revisions, further enhancing the quality of the final manuscript.

**Comment:**

Paper number: essd-2025-425

"DaYu-GCP: All-day Global Cloud Physical Properties dataset with 0.07° resolution retrieved from geostationary satellite imagers covering the period from 2000 to 2022" (Zhao et al) This paper describes a new satellite-based global cloud dataset consisting of cloud phase, top height, optical thickness, and effective radius, called "All-day Global Cloud Physical Properties (DaYu-GCP)", which is designed to overcome limitations in existing cloud data products by providing continuous, global coverage of essential cloud properties from 2000 to 2022, with high spatial and temporal resolution. This was achieved by using a deep learning model, Cloud-SmaAtUNet, which was trained on MODIS and applied to gridded geostationary satellite data (GridSat-B1). The work included evaluation results against MODIS and CALIOP products. The paper presents an ML-based approach to generate consistent global cloud products utilizing existing satellite data, which would potentially provide benefits for climate change research, particularly by enabling long-term, all-day analysis.

Response:

We are grateful for the reviewer's recognition of our research; your assessment has been a significant encouragement to our entire team. We have carefully read and thoroughly considered all of the suggested revisions. With respect to the major comments, we designed and conducted additional experiments, and our conclusions have been further substantiated by the results of these

new analyses. Furthermore, we sincerely appreciate the reviewer's meticulous reading of our manuscript, as evidenced by the numerous minor suggestions for improvement that were provided. This is the first paper authored by the corresponding author. Although the entire research and writing process was carried out with scientific rigor and diligence, some minor issues inevitably remained. These issues have now been fully addressed in response to your comments, as well as those of the other reviewer. As a result, the overall quality of the manuscript has been substantially improved, and we extend our sincere gratitude for your valuable assistance and insights. Your review was exceptionally thorough, and all comments reflect a high level of professionalism.

**Comment 1:**

Additional details and results would be desirable to strengthen the physical interpretation of the model training, particularly given the use of only two IR channels. In particular, for nighttime evaluation, further discussion of cloud optical properties such as cloud optical thickness and effective radius would enhance the scientific robustness of the work for publication.

Response 1:

We are grateful for the reviewers' comments. We fully concur that an in-depth analysis of microphysical properties such as COT and CER would significantly enhance the interpretation of the model's physical mechanisms, particularly during night-time. Following this suggestion, we have strengthened the night-time assessment in the revised manuscript.

The paper employs CALIOP to verify diurnal consistency. Building upon this consistency, annual comparisons with CALIOP were subsequently conducted. Unlike prior studies that treated COT from CALIOP and MODIS as identical products for direct evaluation, we implemented additional screening. Given CALIOP's inability to penetrate thick clouds, this study exclusively utilises CALIOP cloud layers labelled as "transparent" as the true reference for optical thickness. These cloud layers are characterised by laser beams that can fully penetrate to the cloud base, enabling CALIOP to obtain complete and reliable optical thickness measurements. In contrast, opaque cloud layers, in which laser signals undergo complete attenuation within the cloud body, yield no valid cloud base or optical thickness information and were therefore excluded from the assessment.

Additionally, for MODIS we similarly selected regions with thin clouds for evaluation. This approach allows us to exclude COT values from the assessment in cases where CALIOP correctly identified transparent clouds (thin clouds) as thick clouds, whereas MODIS misclassified them. This discrepancy arises from differences in observation principles: MODIS employs passive sensing techniques, whereas CALIOP, as an active lidar system, relies on fundamentally different methodologies. Restricting CALIOP, MODIS, and DaYu-GCP to thin-cloud conditions minimises issues arising from overall inconsistencies between CALIOP and MODIS. Three products underwent additional evaluation and comparison: CLP, CTH, and COT. The COT assessment has been corrected, as illustrated in Fig. 8 (Response figure 1) in the revised manuscript.

[Figure]

**Response figure 1.** The accuracy of the DaYu-GCP (daytime and nighttime) was evaluated using data from the active sensors CALIPSO/CALIOP. (a) CLP accuracy. (b) The RMSE of CTH. (c) The RMSE of COT. The blue bar chart shows the model's daytime accuracy, and the yellow bar chart shows the model's nighttime accuracy.

The absence of CER assessments arises from the lack of CER products in CALIOP for comparison. Existing studies likewise do not provide supplementary night-time evaluations of CER because of this limitation in CALIOP, as illustrated by Table 3 in Zhao et al. (2024a). Nevertheless, owing to its active remote sensing characteristics, CALIPSO/CALIOP data—despite offering higher precision and accuracy than passive remote sensing—are still employed in our evaluations. The absence of CER products in CALIOP reflects a current limitation of observational algorithms, underscoring the need for further investment in research and development of algorithms and products related to cloud physical properties. This also highlights the significance of our research.

Thus, the CALIOP data currently used in the assessment all correspond to thin clouds classified as transparent, while MODIS likewise selects thin clouds with COT values of 5 or less for evaluation. This approach not only makes the assessment more scientifically rigorous but also enhances its reference value.

*"Compared with the active sensors CALIPSO/CALIOP, the Accuracy of CLP during the day and night are 0.787 and 0.775, respectively (Response figure 1 (a)). The daytime and nighttime RMSE of CTH are 3.384 km and 3.568 km, respectively (Response figure 1 (b)). For the evaluation of COT, given that CALIOP cannot penetrate thick cloud layers, this assessment considered only thin cloud layers labeled as transparent in the CALIOP data as ground-truth references. Correspondingly, within DaYu-GCP, only thin cloud regions with a thickness less than 5 were compared against CALIOP measurements. Finally, the daytime and nighttime RMSE of the COT product are 2.836 and 3.002, respectively (Response figure 1 (c)). Overall, the model results demonstrate good consistency between day and night."* **(In section 3.4 Evaluation of DaYu-GCP with CALIOP official products line 323-330)**

**Comment 2:**

The authors present yearly trends of the DaYu-GCP products in Figure 13. To better highlight the strengths of this dataset, it would be beneficial to include an additional comparison with existing global cloud datasets. While comparison with ISCCP or SatCORPS may not be feasible due to the lack of overlapping periods, a comparison with CLARA-A3 or current ABI/AHI products for

specific regions could provide valuable context and validation.

Response 2:

We are grateful for the reviewer's suggestions and consider the proposal for comparative analysis with other global cloud product datasets to be highly valuable. However, because the SatCORPS product commenced in 2023 and differs in temporal coverage from our product, a direct comparison is not feasible. Both ISCCP and CLARA-A3 therefore remain viable candidates for comparison. Nevertheless, a comprehensive comparison would require additional manual processing and corresponding computational adjustments.

The four variables compared in this study are cloud cover frequency (CCF), CTH, CER, and COT.

For ISCCP, the ISCCP basic HGG data used in this study inherently provide global cloud physical properties at a 3 h temporal resolution. The CCF and COT products are directly obtained from the CCF (cldamt) and COT (tau) variables, respectively. In contrast, CTH and CER require further calculations before monthly averages can be derived for statistical analysis. Specifically, CTH is calculated from cloud-top temperature (CTT), while CER is jointly derived from cloud water path (CWP) and COT. Within the ISCCP dataset, CTT and CWP are accessed through the variables tc and wp, respectively.

The CTH calculation follows the guidance provided in the official ISCCP documentation (Schiffer and Rossow, 1983; Rossow et al., 1996; Young et al., 2018). This documentation provides a FORTRAN program, D2READ, whose core principle is to estimate cloud-top height from the difference between CTT and surface temperature, using a fixed temperature lapse rate of $6.5 \text{ K·km}^{-1}$.

CER is calculated using an empirical relationship based on CWP and COT (Liu et al., 2025). For water clouds, CWP represents the liquid water path (LWP), whereas for ice clouds it represents the ice water path (IWP). The calculation formulas for LWP and IWP differ slightly, and the official ISCCP documentation distinguishes between them according to CTT (tc). Clouds with CTT < 260 K are classified as ice clouds, while those with CTT > 260 K are classified as water clouds. Accordingly, CER for water clouds (Nakajima and Nakajma, 1995) and ice clouds (Liou, 2002) is calculated using the following formula:

$$CER = \begin{cases} \frac{3}{2} \cdot \frac{LWP}{\rho \cdot COT}, & if \ CTT < 260K \\ \frac{b}{2(\frac{COT}{IWP} - c)}, & if \ CTT > 260K \end{cases} \quad (1)$$

Where $\rho = 1.00 \text{ kg·m}^{-3}$ is the density of liquid water. $b = -6.565 \times 10^{-3}$ and $c = 3.686$ are the coefficients used to calculate the CER for ice clouds.

Secondly, for CLARA-A3, the specific data ID used during download is EO:EUM:DAT:0874, which requires downloading four products: CPH, CTO, IWP, and LWP. CCF is obtained by directly reading the cph variable from the CPH product, while CTH is obtained by directly reading the cth variable from the CTO product. COT is calculated from the cot_ice and cot_liq variables in the IWP and LWP products using the following formula (Nakajima and King, 1990):

$$COT = cot\_ice + cot\_liq \quad (2)$$

Where cot_ice denotes the COT of ice clouds, and cot_liq denotes the COT of water clouds.

CER is calculated from the iwp, lwp, cre_ice and cre_liq variables of IWP and LWP respectively, using the following formula (Nakajima and King, 1990):

$$CER = \frac{cre\_liq \times lwp + cre\_ice \times iwp}{lwp + iwp} \quad (3)$$

Where cre_ice denotes the CER for ice clouds, cre_liq denotes the CER for water clouds, and iwp and lwp denote the WCP for ice clouds and water clouds respectively.

We processed the ISCCP and CLARA-A3 data according to the aforementioned formula and compared the results with our DaYu-GCP. To more clearly identify the annual trends, we plotted the anomaly for each variable. The anomaly was obtained by subtracting the long-term climatological mean from each numerical value. The results are presented in Fig. 11 of the manuscript. For ease of reference, we have also included the figure below, labeled as Response figure 2.

[Figure]

**Response figure 2.** CCF, CTH, COT and CER anomaly change curves for the period 2000 to 2022 across the 60°N to 60°S latitude zone, derived from DaYu-GCP, ISCCP and CLARA-A3 products. Statistical data are presented separately for the Northern and Southern Hemispheres. The black, red and blue curves represent DaYu-GCP, ISCCP and CLARA-A3 respectively. In the corner, we have labelled the Pearson correlation coefficients $R_{ISCCP}$ and $R_{CLARA-A3}$. $R_{ISCCP}$ denotes the correlation coefficient between DaYu-GCP and ISCCP. $R_{CLARA-A3}$ denotes the correlation coefficient between DaYu-GCP and CLARA-A3. (a) and (e) are CCF variables. (b) and (f) are CTH variables. (c) and (g) are CTH variables. (d) and (h) are CTH variables.

*To clearly compare the annual variability of cloud physical properties among different products, monthly anomaly time series of each parameter were constructed for the Northern and Southern Hemispheres based on the monthly mean data from each product, and Pearson correlation coefficients (R) between DaYu-GCP and the other two datasets were calculated (Response figure 2). DaYu-GCP exhibits higher correlations with ISCCP in CCF, COT, and CER; for the Northern Hemisphere, the R reach 0.760, 0.764, and 0.514, respectively. DaYu-GCP shows higher correlation with CLARA-A3 in CTH, with R = 0.778 for the Northern Hemisphere. The Southern Hemisphere exhibits similar results. Among all R values, CER shows the lowest correlation compared to CCF*

*and CTH; for the Northern Hemisphere, R values between DaYu-GCP and ISCCP, DaYu-GCP and CLARA-A3, and ISCCP and CLARA-A3 are 0.514, 0.412, and 0.219, respectively. This indicates that CER remains a challenging parameter in future cloud physical properties datasets. These comparison results demonstrate that, as a new cloud product, DaYu-GCP exhibits reasonable consistency in the spatiotemporal variability of major cloud parameters with the two long-standing global datasets, ISCCP and CLARA-A3. This underscores the reliable data quality of DaYu-GCP and its potential as a consistent data source for global cloud-climate studies.* **(In section 3.4 Spatiotemporal distribution characteristics of clouds line 384-395)**

**Comment 3:**

Major comments

MODIS L2 cloud products have been used in the initial Cloud-SmaAtUNet model training, which would be anticipated to produce degraded cloud optical properties at night due to lack of visible information. Although the authors designed DaYu-GCP generating day and night cloud products utilizing both GridSat and ERA-5 and show consistent trends against MODIS and CALIOP (CLP and COT), it does not provide clear explanations about using two IR bands only and estimation of nighttime evaluation results, especially for comparisons with CALIOP data (such as cloud optical thickness and effective radius). More discussions about how to match with MODIS and CALIOP products and physical explanations about nighttime comparisons (1-2 additional figures) would be needed.

Response 3.1:

**We are grateful for the reviewer's comments. In response to this particular point, we have provided separate replies under Responses 3.1, 3.2, and 3.3.**

Firstly, regarding Response 3.1, the reason for utilising only two infrared bands is that the GridSat-B1 dataset provides information from three channels: the visible channel (0.6 μm), the infrared water vapour (IRWVP) channel (6.7 μm), and the CDR-quality infrared window (IRWIN) channel (11 μm). As our objective is for DaYu-GCP to exhibit all-day capability, the visible channel becomes unavailable during night-time conditions. This limitation renders the retrieval of night-time cloud physical property products infeasible and impractical. Conventional cloud physical property products either exclude night-time data entirely (e.g., Himawari and CARE, as described in the manuscript introduction) or apply separate algorithms for daytime and night-time conditions (e.g., GOES ABI), which can lead to discontinuities or systematic biases during transition periods. Therefore, using the IRWVP and IRWIN infrared channels for training and inversion allows the generation of cloud physical property products that cover all times of day (both daytime and night-time), while also ensuring consistency between daytime and night-time retrievals.

Having clarified the technical implementation, readers and reviewers may naturally question whether the use of only two infrared bands can guarantee sufficient retrieval accuracy and whether this choice has a sound physical basis. Accordingly, the subsequent work in our manuscript focuses on ensuring the reliability of the retrievals and the accuracy of the DaYu-GCP dataset. First, evaluation of the Cloud-SmaAtUNet retrieval results on the test set demonstrates that, despite using fewer brightness temperature channels, the model performance is comparable to the current state of the art in the field. This performance can be attributed to the extended temporal coverage of the

input data, the sufficiently large sample size, and the adaptability of the Cloud-SmaAtUNet model. Subsequently, we conducted further assessments of the generated DaYu-GCP dataset. These included an evaluation of annual temporal consistency against MODIS, an assessment of spatial consistency, and the use of CALIPSO/CALIOP to further examine diurnal consistency and interannual temporal consistency.

The physical mechanism underlying the use of two infrared spectral regions was not described in the original manuscript. Following the joint recommendations of Reviewer 1 and Reviewer 2, we have supplemented the study with additional experiments to clarify the physical interpretation. In total, three sets of experiments were conducted:

Experiment 1:

Employing the pixel-based (point-to-point) single-layer cloud retrieval machine learning model Random Forest (Cloud-RF) as a comparative benchmark. Both Cloud-SmaAtUNet and Cloud-RF utilise the complete input dataset as model inputs. Scatter plots were employed to compare the training outcomes, thereby further elucidating the physical characteristics of the models.

Experiment 2:

Additionally, for both the Cloud-SmaAtUNet and Cloud-RF models, we configured three distinct input sets. These comprised: ① data including both 11 μm and 6.7 μm brightness temperatures together with the full ERA5 meteorological dataset; ② data consisting solely of 11 μm and 6.7 μm brightness temperatures; and ③ data consisting solely of ERA5 meteorological fields. This experiment employed both the Cloud-SmaAtUNet and Cloud-RF models for training and evaluation, enabling a comparison of the effects of different input configurations on retrieval accuracy. It also reveals the respective contributions of the 11 μm and 6.7 μm brightness temperature data and the ERA5 meteorological fields to the retrieval results.

Experiment 3:

Using radiative transfer calculations of brightness temperature data alongside corresponding cloud product data, this experiment aims to quantitatively assess the noise level of GridSat brightness temperature data through numerical simulation methods.

The study employed the integrated, high-efficiency radiative transfer model DaYu. By inputting ERA5 meteorological fields and cloud products into the DaYu model, corresponding brightness temperature data can be generated. Detailed specifications of the radiative transfer model are described in Li et al. (2023b). In brief, it employs the Optimized alternate Mapping Correlated K–Distribution (OMCKD) gas absorption scheme, combined with cloud and aerosol optical parameterizations, and uses a 2N-Discrete Ordinate aDding Approximation (DDA) radiative transfer solver. By integrating input data from ERA5 and MODIS, it enables efficient and high-precision simulation of reflectance and brightness temperature for Himawari-8/AHI satellite imagery (Li et al., 2023b).

During training, Gaussian noise was progressively added to the brightness temperature data generated by the radiative transfer model, with noise intensities of $\sigma = 0.25, 0.5, 0.75$, and 1 K. By statistically analyzing the retrieval errors of cloud products (CLP, CTH, COT, CER) under these varying noise levels, we quantified the noise characteristics of the GridSat-B1 data used in DaYu-GCP.

The implementation details and results of these experiments have been incorporated and

revised in the manuscript body and references. The outcomes of several experiments are presented below:

Results of Experiment 1:

These results are presented in the form of a scatter plot, as shown in Response figure 3. This figure replaces Figure 3 in the main text, with the left column displaying the content of Figure 3 from the original submitted manuscript. **It should be clarified that, since the coefficient of determination ($R^2$) and Pearson correlation coefficient (Pearson R) serve equivalent functions, comparing these two distinct metrics may easily cause confusion. As Pearson R is more commonly used in similar studies, $R^2$ has been omitted. Furthermore, for scatter plots of COT, using linear coordinates would cause most data points (in the low-value range) to cluster near the bottom, making it difficult to discern the distribution details, while the few high-value points could distort the visual balance. Consequently, logarithmic scaling was applied throughout the plotting process.** To enhance visual clarity, we have adjusted the color bar. We also compared the performance of the image-based Cloud-SmaAtUNet with that of the pixel-based (point-to-point) Cloud-RF using identical input data, including 6.7 μm and 11 μm brightness temperatures and ERA5 meteorological fields.

*Compared with the MODIS official products, Cloud-SmaAtUNet achieved an overall accuracy of 82.3% in the CLP classification task. The recognition accuracies of Cloud-SmaAtUNet for clear sky, water clouds, and ice clouds reached 81.52%, 80.21%, and 86.38%, respectively (Response figure 3 (a)). In addition, Cloud-SmaAtUNet also exhibited relatively dense joint probability density distributions aligned along the diagonal in the regression tasks for CTH, COT, and CER. Although Cloud-SmaAtUNet showed a systematic underestimation (MBE < 0), the estimation errors for CTH, COT, and CER remained within acceptable ranges (Response figure 3 (c)(e)(g)). The RMSE (MAE) values were 1.617 km (0.954 km), 11.314 (6.871), and 7.181 μm (5.133 μm), respectively, with PearsonR of 0.926, 0.617, and 0.799. These results indicate that Cloud-SmaAtUNet exhibits excellent performance in the retrieval of cloud physical properties.*

*In contrast, the performance of the Cloud-RF model in cloud retrieval declined across all products. The overall accuracy of Cloud-RF in the CLP classification task decreased to 78.1% (Response figure 3 (b)). Among the classes, the recognition accuracy for water clouds decreased most markedly (73.44%), while those for clear sky and ice clouds also showed slight reductions (78.93% and 85.19%, respectively). This indicates that Cloud-RF performs worse than Cloud-SmaAtUNet in CLP classification. Moreover, in the regression tasks for CTH, COT, and CER, Cloud-RF exhibited more pronounced underestimation than Cloud-SmaAtUNet (MBE < 0 with larger magnitude), and the joint probability density distributions aligned along the diagonal were less concentrated (Response figure 3 (d)(f)(h)). The RMSE (MAE) values were 2.369 km (1.483 km), 13.370 (8.935), and 8.860 μm (6.535 μm), respectively, with PearsonR of 0.843, 0.565, and 0.719. These results demonstrate that, compared with Cloud-SmaAtUNet, the Cloud-RF model shows inferior performance in retrieving cloud physical properties.*

*Compared with the conventional Cloud-RF, Cloud-SmaAtUNet can leverage the spatial structure information of clouds to improve cloud physical properties retrievals. The accuracy of CLP is increased by 5.4%, while the RMSE of CTH, COT, and CER are reduced by 31.7%, 18.2%, and 23.4%, respectively. Beyond the performance differences, Cloud-SmaAtUNet requires only*

*about 100 s to retrieve a single global image, whereas Cloud-RF takes nearly six times longer, approximately 630 s. These findings are also reflected in other studies (Zhao et al., 2024a).* **(In section 3.2 Physical interpretability of the Cloud-SmaAtUNet model line 220-245)**

Experimental conclusions: Cloud-RF employs a pixel-based point-to-point learning approach, thereby lacking effective capture of spatial contextual features. In contrast, Cloud-SmaAtUNet, as an image-based learning algorithm, demonstrates significant advantages in extracting spatial characteristics of cloud physical products. The results of this experiment confirm Cloud-SmaAtUNet's superiority in this regard.

[Figure]

**Response figure 3.** Detailed distribution of testing-set evaluations on Cloud-SmaAtUNet (left column) and Cloud-RF (right column). (a-b) Confusion matrix for CLP. Scatter density distribution diagram under the kernel density estimation of (c-d) CTH. (e-f) COT. (g-h) CER between MODIS official and DaYu-CLAS Cloud-SmaAtUNet or DaYu-CLAS Cloud-RF products.

Results of Experiment 2:

This experiment comprised three groups in total:

Group 1: Brightness temperature (BT) at 11μm and 6.7μm with ERA5 meteorological fields

Group 2: 11μm and 6.7μm BT

Group 3: ERA5 meteorological fields

Results are presented in Response table 1 and Response figure 4, with supplementary details included in the manuscript (corresponding images are provided in Table S1 and Figure 2 of the Supplementary Materials). *The CLP classification results in Response table 1 indicate that when using only ERA5 data, the Cloud-SmaAtUNet classification accuracy for clear sky, water cloud, and ice cloud was 66.06%, 63.64%, and 68.16%, respectively; when using only brightness temperature (BT) data, the accuracy increased to 78.86%, 79.43%, and 83.96%. When BT and ERA5 were combined, the classification accuracy for clear sky, water cloud, and ice cloud further improved to 81.52%, 80.21%, and 86.38%, representing increases of 23.4%, 26.0%, and 26.7% compared with using ERA5 alone, and increases of 3.4%, 1.0%, and 2.9% compared with using BT alone. The same conclusions apply to Cloud-RF. When only ERA5 data were used, the classification accuracy of Cloud-RF for clear sky, water cloud, and ice cloud was 61.56%, 59.42%, and 63.72%, respectively. When only BT data were used, the corresponding accuracy increased to 73.52%, 72.05%, and 82.77%. When both BT and ERA5 were combined, the classification accuracy for clear sky, water cloud, and ice cloud further increased to 78.93%, 73.44%, and 85.19%, representing improvements of 28.2%, 23.6%, and 33.7% compared with using ERA5 alone, and improvements of 7.4%, 1.9%, and 2.9% compared with using BT alone. These results confirm that meteorological fields primarily describe the environmental potential for cloud formation and evolution rather than providing direct observational signals of clouds, and therefore cannot independently achieve high-accuracy instantaneous cloud state identification. When only dual-channel brightness temperature data are used, model performance improves because these data directly capture the radiative characteristics of cloud tops. However, the combination of brightness temperature and ERA5 meteorological fields yields optimal performance. This highlights a key mechanism: meteorological fields, as continuous atmospheric background fields, provide causal interpretation and physical constraints for the cloud signals observed in brightness temperature data.*

**Response table 1.** Cloud-SmaAtUNet and Cloud-RF cloud classification retrieval accuracy under different input groups.

| Group | Cloud-SmaAtUNet (%) | | | Cloud-RF (%) | | |
|---|---|---|---|---|---|---|
| | Clear sky | Water cloud | Ice cloud | Clear sky | Water cloud | Ice cloud |
| 1 | 81.52 | 80.21 | 86.38 | 78.93 | 73.44 | 85.19 |
| 2 | 78.86 | 79.43 | 83.96 | 73.52 | 72.05 | 82.77 |
| 3 | 66.06 | 63.64 | 68.16 | 61.56 | 59.42 | 63.72 |

Here, we have statistically analyzed the RMSE for CTH, COT, and CER cloud products derived from the Cloud-SmaAtUNet retrieval, categorized by value range, as shown in Response figure 4. For example, for CTH (see Response figure 4 (a)), RMSE was calculated for the intervals 0–2, 2–4, 4–6, 6–8, 8–10, 10–12, 12–14, 14–16, and 16–18. As CTH values do not exceed 18, no

interval above 18 was defined. Moreover, the division of value ranges is not uniform. For instance, in the case of COT (see Response figure 4 (b)), the initial intervals remain 2 units wide, but subsequent intervals progressively increase, culminating in the 50–60 and 60+ ranges. The choice of interval divisions is primarily determined by the characteristics of the data.

*The results of the regression tasks are shown in Fig. 4. When using ERA5 alone, the RMSE of CTH, COT, and CER retrieved by Cloud-SmaAtUNet across different ranges were 2.1–7.2 km, 12.9–82.9, and 7.0–25.3 μm, respectively. When using only BT, the RMSE was 1.5–6.0 km, 9.2–63.7, and 4.8–24.1 μm. When using BT and ERA5 together, the RMSE of CTH, COT, and CER decreased to 1.1–5.1 km, 6.0–45.9, and 2.8–21.4 μm. These results indicate that using BT and ERA5 together, compared to using only BT (or only ERA5), reduces the RMSE of CTH, COT, and CER on average by 21.0% (63.5%), 39.5% (90.3%), and 42.6% (85.6%), respectively. The above experimental results indicate that satellite-observed BT represents an integrated signature of clouds, the atmosphere, and surface types. In the cloud retrieval process based on BT, incorporating atmospheric background fields and surface information contributes to the accurate retrieval of cloud physical properties.*

[Figure]

(a) CTH

(b) COT

(c) CER

**Response figure 4.** Cloud-SmaAtUNet (left column) and Cloud-RF (right column) were evaluated for RMSE across three distinct input datasets, with results presented for different value ranges. Group 1 inputs comprise 11 μm and 6.7 μm BT alongside ERA5. Group 2 inputs comprise 11 μm and 6.7 μm BT. Group 3 inputs comprise ERA5. (a) CTH. (b) COT. (c) CER.

**(In section 3.2 Physical interpretability of the Cloud-SmaAtUNet model line 250-275)**

Results of Experiment 3:

The core design of this experiment is as follows:

First, an ideal training dataset was constructed: we employed the integrated and efficient radiative transfer model DaYu, using high-precision ERA5 reanalysis data (providing meteorological fields such as temperature and humidity) and the official MODIS cloud products COT and CER as inputs to simulate the corresponding 6.7 and 11 μm satellite channel brightness temperatures (BTs). This simulation process ensures physical consistency and eliminates

observational errors. Throughout this process, the simulated brightness temperatures remain physically consistent with the input meteorological fields and cloud products, forming an ideal benchmark dataset that is readily suitable for model training.

Secondly, utilizing the aforementioned observation-error-free, physically consistent 'simulated BT cloud product' dataset, a model was trained with Cloud-SmaAtUNet to invert cloud physical products from 6.7 and 11 μm brightness temperatures combined with meteorological fields. The performance of the model trained under these ideal conditions represents the theoretical upper limit of retrieval algorithms when free from observational error interference.

Finally, controlled noise was introduced for sensitivity experiments: to investigate the effects of noise on inversion and assess the actual noise level of GridSat-B1 data, Gaussian white noise of varying intensities was sequentially added to the simulated 'ideal' BT data. Noise standard deviations (σ) were set to 0.25, 0.5, 0.75, and 1.0 K, respectively. Subsequently, cloud products were re-inverted using these noisy brightness temperature data. By analyzing the degradation pattern of retrieval accuracy with increasing noise levels, a quantitative relationship between noise intensity and retrieval error was established. Finally, by substituting the retrieval error of the DaYu-GCP, established using GridSat-B1 brightness temperatures, into this relationship, we can infer the approximate range of equivalent noise for GridSat-B1 BT data.

[Figure]

**Response figure 5.** The accuracy and RMSE of cloud products retrieved from radiative transfer model calculations incorporating Gaussian white noise of varying intensities. The STD σ of the noise was set to 0.25, 0.5, 0.75, and 1.0K respectively. The blue line represents the retrieval accuracy and RMSE of cloud products at these noise levels. The green line indicates the accuracy and RMSE of the DaYu-GCP cloud product. (a) CLP accuracy. (b) CTH RMSE. (c) COT RMSE. (d) CER RMSE.

Comparing retrieval results under different noise levels with DaYu-GCP shows that the equivalent noise interval estimates for GridSat-B1 brightness temperature data are presented in

Response figure 5. Among these, CLP retrieval is evaluated using accuracy, while the other products—CTH, COT, and CER—are assessed using RMSE. The equivalent noise of GridSat-B1 brightness temperature in CLP retrieval is approximately 0.37 K, in CTH it is about 0.38 K, in COT around 0.39 K, and in CER approximately 0.37 K. This experiment indicates that the GridSat-B1 brightness temperature used in DaYu-GCP exhibits equivalent noise comparable to that of simulated data with σ = 0.35–0.4 K. Overall, this experiment demonstrates that although GridSat-B1 is a global brightness temperature product derived from multi-satellite observations, its infrared noise-equivalent temperature difference remains within an acceptable range.

Response 3.2:
Response 3.2 addresses the issue by outlining the methodology for estimating MODIS and CALIOP product matching during nighttime assessments, along with a comparative analysis between DaYu-GCP and CALIOP data.

Regarding data matching, we address below the temporal and spatial matching strategies, as well as how we handle the issue of horizontal homogeneity in the matching process.

Temporal matching was conducted according to the specific temporal characteristics of each dataset. MODIS and ERA5 data, available at synoptic hours (0000, 0300, …, 2100 UTC), were directly matched. In contrast, CALIOP data were included by selecting profiles within a ±2-minute window around each corresponding synoptic hour.

Section 2.2 of the manuscript, titled "2.2 Method," provides a detailed description of the time-matching constraints.

*"To ensure the accuracy of the evaluation, the time difference between CALIOP and DaYu-GCP was limited to ±2 minutes. The time difference between CALIOP and MODIS was also limited to ±2 minutes."* **(In section 2.2 Method line 180)**

Secondly, regarding spatial scale:
This study employed resampling methods tailored to the characteristics of each dataset. For the MODIS Level-2 cloud product (MOD06/MYD06), the nearest-neighbour interpolation method was applied. This approach is particularly suitable for categorical or discrete data products, as it effectively preserves the integrity of the original pixel values (Huang et al., 2012). For the ERA5 meteorological reanalysis fields, bilinear interpolation was used. This method is appropriate for continuous data fields, as it maintains spatial gradients while producing a smooth transition between values (Kim et al., 2019).

The 1-km cloud product from the CALIOP sensor onboard the CALIPSO satellite is recorded as geolocated data points with corresponding values. Although CALIOP observes and records the full vertical cloud phase profile, the dominant phase at the cloud top is used as the representative value for each CALIOP profile in our processing (Li et al., 2023a; Zhao et al., 2024a). The data processing strategy differed slightly between the categorical product (CLP) and the continuous products (CTH, COT, CER). All products were first remapped onto a regular 0.01° latitude–longitude grid. Subsequently, to align with the study's target 0.07° grid, specific aggregation rules were applied.

For the categorical CLP product, the value assigned to each target grid cell was determined by the mode—the most frequently occurring value among the underlying 0.01° grid cells within the

target cell. For the continuous products (CTH, COT, CER), the value assigned was the arithmetic mean, calculated exclusively from the 0.01° grid cells that had been classified, via the aforementioned method, into the predominant cloud type (CLP value) of the target grid cell.

Here is a simple example: if a target 0.07° grid cell contains 11 underlying 0.01° grid cells with CLP values of 0 (clear sky) occurring 3 times, 1 (water cloud) 2 times, and 2 (ice cloud) 6 times, the mode is 2 (ice cloud). Thus, the target cell's CLP value is assigned as 2. Subsequently, for the CTH, COT, and CER products, the arithmetic mean is computed using only the 6 values from the cells where CLP = 2. This mean value is then assigned to the target grid cell for each respective continuous product.

This two-step aggregation strategy not only ensures spatial consistency between the discrete classification (CLP) and the continuous retrievals (CTH, COT, CER) at the target resolution but also physically accounts for the distinct vertical stratification of ice clouds (typically high-altitude) and water clouds (typically low-altitude), thereby reasonably mitigating the issue of horizontal homogeneity within the grid cell. **(In section 2.2 Method)**

This description has also been incorporated into Section 2.2 (method part) of the manuscript.

Response 3.3:

Response 3.3 primarily explains the physical mechanism for retrieving night-time cloud optical properties using only infrared bands.

Despite the absence of visible-light bands at night, DaYu-GCP can still estimate COT and CER through the synergistic use of two infrared bands—11 µm IRWIN (BT11) and 6.7 µm IRWVP (BT6.7)—together with ERA5 meteorological fields. The physical interpretation is grounded in the radiative transfer characteristics of clouds within the thermal infrared spectrum (Liou, 2002):

The IRWIN channel lies within the atmospheric window and is highly sensitive to both surface and cloud-top temperatures. BT11 is directly related to cloud-top height (CTH) and cloud-top temperature. For optically thick clouds, BT11 approximates the physical temperature at the cloud top; for thin cirrus clouds, BT11 represents a composite signal of cloud emission and the underlying atmospheric or surface emission.

The IRWVP channel lies within the water-vapour absorption band and is sensitive to mid-to-upper atmospheric water content. The cloud brightness temperature (BT6.7) in this channel is influenced by both cloud height and the upper-level water vapour. The brightness temperature difference between BT11 and BT6.7 (BTD11-6.7) is a critical parameter: for high-level clouds (e.g., cirrus) located above the water-vapour absorption layer, BT6.7 is minimally affected by water vapour, yielding a small (or even negative) BTD11-6.7 value. For mid-to-low-level water clouds, positioned below or within the water-vapour layer, the BT6.7 signal is attenuated by upper-level water vapour, resulting in a larger BTD11-6.7 value.

Therefore, BTD11-6.7 exhibits a strong correlation with cloud-top pressure/altitude, providing a physical basis for distinguishing cloud phases and estimating cloud height. Moreover, the temperature and humidity profiles supplied by ERA5 offer a physically constrained context for interpreting these brightness temperature signals, thereby better constraining CTH.

At night, for water clouds, COT and CER are related to the cloud's emissivity at 11 µm (derivable from the cloud-top temperature estimated via BT11 and ERA5 profiles) and to the cloud microphysical model. Although infrared sensitivity to COT saturates at high optical thicknesses

(typically COT > 5), estimation remains feasible for most thin clouds where COT < 5. Specifically: BT11 observations combined with ERA5 temperature profiles → inference of cloud infrared emissivity ε → using the theoretical relationship between ε and COT, estimation of COT.

CER estimation relies more heavily on cloud height and phase-state information constrained by BTD11-6.7 and ERA5 profiles, combined with cloud microphysical characteristics learned from extensive training data (e.g., different CER distributions for water and ice clouds).

Our deep learning model (Cloud-SmaAtUNet) implicitly learns this complex, non-linear CER–COT mapping, including both daytime and night-time modes, from vast amounts of paired [BT11, BT6.7, ERA5 meteorological fields] → [MODIS cloud products] data, enabling accurate retrieval.

Finally, we have enhanced the night-time contrast analysis using CALIOP, with the revised results presented in Figure 9 of the manuscript. Data processing and interpolation during evaluation follow the approach described in Response 3.2, while the results analysis is demonstrated in Response 1. These modifications collectively demonstrate that DaYu-GCP provides a physically consistent, spatio-temporally continuous, day–night, all-coverage cloud product, effectively addressing gaps or deficiencies in existing products during night-time periods.

**Comment 4:**

Minor comments / typos
Section 2.2.1 and 2.2.2 have several overlaps, which could be better reorganized.
Response 4:
Here, I would first like to express my sincere gratitude to the reviewers for their patient reading and for their tolerance of the minor errors in our manuscript. This is the first research paper for which I am the first author. Although I maintained patience and rigor throughout the experimental process and the preparation of the manuscript, numerous minor issues did persist. I sincerely appreciate the reviewers' meticulous scrutiny of the manuscript and their constructive suggestions for refinement. Each minor point has been carefully considered and addressed, with specific responses provided under the relevant comments. Resolving these issues has markedly enhanced the overall quality of the paper, and I am profoundly grateful for the reviewers' invaluable assistance throughout this process.

We are grateful to the reviewers for their comments on this section and have incorporated the relevant revisions. This section did contain repetition and redundancy, and it has therefore been rephrased. Firstly, Sections 2.2.1 and 2.2.2 have been merged and consolidated into Section 2.2 Method.

Regarding the introduction to the method: this section begins by detailing the models employed in the study. We describe their origin: they are derived from our single-layer and double-layer cloud retrieval models within the DaYu Cloud Analysis System (DaYu-CLAS), with our research utilizing the single-layer cloud retrieval model Cloud-SmaAtUNet. This model demonstrates high accuracy and efficiency in cloud retrieval tasks. It has been applied to H8-AHI retrieval and produced one year of regional cloud products (2017). However, current research has not yet achieved global cloud physical property products, nor does it provide sufficiently long-term temporal coverage.

Subsequently, the production workflow for DaYu-GCP is outlined, encompassing spatiotemporal matching strategies for MODIS official, CALIOP official, and ERA5 data. A

processing strategy is detailed for GridSat-B1 imagery, which is too large for direct input (cropped into 64 × 64 pixel sub-samples), alongside model inputs, training objectives, and validation of the final DaYu-GCP output.

The third section details the dataset sample size and the division of training and test sets. The fourth section outlines the primary model training parameters and the metrics employed for evaluation.

The revised content of Section 2.2 in the manuscript is as follows:
"

**2.2 Method**

*In our previous studies, the single-layer and double-layer cloud retrieval models within the DaYu CLoud Analysis System (DaYu-CLAS) have been demonstrated to perform well in all-day cloud physical properties retrievals. DaYu-CLAS includes single-layer cloud retrieval models such as Cloud-ResUNet (Zhao et al., 2023; Zhao et al., 2024b; Tong et al., 2023), Cloud-SmaAtUNet (Li et al., 2023a), and the CloudDiff model (Xiao et al., 2025), as well as the double-layer cloud retrieval model OverlapCloudDiff (Li et al., 2025). Cloud-SmaAtUNet is an improved version of UNet, in which depthwise separable convolutions and convolutional block attention modules (CBAM) are integrated into both the encoder and decoder paths. Li et al. (2023a) applied Cloud-SmaAtUNet to H8-AHI data and demonstrated that Cloud-SmaAtUNet achieves high accuracy and efficiency in cloud physical properties retrieval tasks. However, that study focused on a single sensor and produced cloud products for only one year (2017), which is insufficient to support studies of global cloud physical properties at high spatiotemporal resolution.*

*Therefore, in this study, the Cloud-SmaAtUNet model is applied to achieve global all-day cloud physical properties retrievals, as shown in Response figure 6. Due to the different institutional sources of the datasets, they may have different projection methods and spatiotemporal resolutions. To ensure the correct correspondence of pixels and data consistency, the data were first aligned to a unified 0.07° latitude and longitude grid before model construction. Temporal matching was conducted based on the respective temporal characteristics of the datasets. MODIS and ERA5 data, available at synoptic hours (0000, 0300, ..., 2100 UTC), were directly matched. In contrast, CALIOP data, were included by selecting profiles within a ±2 min window around each corresponding synoptic hour. Because the GridSat-B1 images are too large to be directly used as model inputs, each image is divided into multiple 64 × 64 pixel sub-images. After data preprocessing, Cloud-SmaAtUNet is trained using brightness temperature (BT) from the 6.7 and 11 μm channels and the SAZ as the primary inputs, with CLP, CTH, COT, and CER from the MODIS official cloud products as labels. Considering the influence of meteorological conditions on cloud formation and development, additional meteorological variables, such as temperature and humidity profiles, are incorporated as auxiliary inputs. In this way, a DaYu-GCP dataset with a temporal resolution of 3 h and a spatial resolution of 0.07° is retrieved. Finally, the products are validated using the MODIS and CALIOP official cloud products to evaluate their spatiotemporal continuity and day–night consistency.*

[Figure]

**Response figure 6.** Flowchart of DaYu-GCP production. It is worth noting that the input data consists of 19 channels, including GridSat-B1 and ERA5, while the target is CLP, CTH, COT, and CER images from MODIS official product processed to the same resolution. The training input and output image sizes are both 64×64. For each target, this study conducted model training and validation and evaluation separately. Finally, a sliding window fusion strategy was used to produce global-scale cloud product data.

*During model parameter training, the spatiotemporally matched dataset comprised approximately 700,000 sample pairs. Data spanning 2000–2021 were used as the training set, while the entire dataset from 2022 constituted the testing set. These samples were evenly distributed across the spatial domain, and the strategy for selecting the training and testing sets satisfied the basic requirements of Cloud-SmaAtUNet. This selection strategy, together with the large overall sample size, jointly reduces the risk of model overfitting. To further expand the training data, data augmentation operations—including horizontal flipping, vertical flipping, and rotations of 90°, 180°, and 270°—were applied to the training set, increasing its size to six times the original. Meanwhile, the testing set remained unchanged, without any augmentation, in order to objectively evaluate the effectiveness of data augmentation in improving model performance. Indeed, the application of data augmentation led to improved accuracy. Detailed information on the dataset is provided in Response table 2.*

**Response table 2.** Sample size and division of the matched dataset into training-set and testing-set.

| Data matching pair | Total number of samples | Training -set | Testing- set | Training-set (data augmentation) | Testing-set (data augmentation) |
|---|---|---|---|---|---|
| GridSat-B1 & MOD06 Labels | 373269 | 357744 | 15525 | 2146464 | 15525 |
| GridSat-B1 & MYD06 Labels | 324950 | 312090 | 12860 | 1872540 | 12860 |
| Combined MOD06 & MYD06 Labels | 698219 | 669834 | 28385 | 4019004 | 28385 |

*Key parameters for model training include: batch size = 512, maximum epochs = 300, and learning rate = 0.001. An early-stopping strategy was adopted, whereby training was terminated if the loss on the testing set did not decrease by more than 0.1 for 15 consecutive epochs. All models converged and stopped before reaching the maximum of 300 epochs. The loss functions used for model training varied by task. CrossEntropyLoss was employed for the CLP classification task, whereas MSELoss was applied to the CTH, COT, and CER regression tasks. To achieve seamless global coverage of the cloud products, an image sliding-window fusion strategy was implemented to eliminate gaps between adjacent small samples; details of this method are provided in Supplementary Material Text S1. For model evaluation, statistical metrics for classification performance included Accuracy, Recall, Precision, and F1-score, while regression performance was assessed using root mean squared error (RMSE), mean absolute error (MAE), mean bias error (MBE), and Pearson correlation coefficients (PearsonR).*

" **(In section 2.2 Method line 164-215)**

**Comment 5:**

Section 5 Data availability could be placed after Section 6.

Response 5:

We are grateful to the reviewers for their careful reading and constructive feedback. It is indeed true that placing the data availability section before the conclusions disrupts the paper's logical flow and structure. However, due to the specific requirements of the ESSD journal, all published papers must explicitly state data availability within the main text. Consequently, we have not removed this section numbering; instead, we have reclassified data availability as Section 6 and moved the conclusions to Section 5. As a result of this reorganization, the current section numbers have been shifted: Section 4 now presents the conclusions, and Section 5 covers data availability.

**Comment 6:**

Line 57: "Wang et al 2024 -> Better references such as Bessho et al 2016

Response 6:

We thank the reviewers for their comments. Indeed, as you point out, Bessho et al. 2016 is a more appropriate reference for introducing Himawari-8, as it was the original paper presenting Himawari-8/9. We have read and amended our citation to include:

An Introduction to Himawari-8/9—Japan's New-Generation Geostationary Meteorological Satellites.

Additionally, concerning the AHI official cloud product, we have incorporated the paper 'Cloud Property Retrieval from Multiband Infrared Measurements by Himawari-8' by Iwabuchi et al., 2018.

The relevant sections of the manuscript have now been revised accordingly:

*"Advanced Himawari Imager (AHI) onboard the Himawari-8/9 (H8/9) satellites operated by the Japan Aerospace Exploration Agency (JAXA), can monitor East Asia and the Pacific region (Bessho et al., 2016b, a; Iwabuchi et al., 2018)."* **(In section 1 Introduction line 48-49)**

**Comment 7:**

Line 84-86: Complete the sentence "For example, calibrating raw sensor signals to radiant …"

Response 7:

The issue with this sentence is the absence of a subject, rendering it incomplete. This resulted from an oversight during the drafting process. We have now amended the sentence to ensure it is grammatically complete and accurately conveys the intended meaning.

"For example, generating consistent products entails calibrating raw sensor signals to radiant brightness (Helder et al., 2020; Lee et al., 2024) and brightness temperature, and accurately mapping each pixel to Earth surface coordinates (Knapp et al., 2011; Jiao et al., 2024)."

**Comment 8:**

Line 99: "in the fact that DaYu-GCP is the first global dataset with a spatial resolution of 0.07° and a temporal resolution of 3 h." -> The statement would be better revised with additional specifications regarding what distinguishes this dataset. It may be clarified whether the emphasis lies on its comparatively high spatial and temporal resolutions relative to existing global cloud datasets such as ISCCP, NASA's SatCORPS, and CLARA-A3, for instance, if the authors intended. Also, specifying that it is a "satellite-based" global cloud dataset would provide important context of the description.

Response 8:

We are sincerely grateful for the reviewer's valuable comments and have made revisions in accordance with your suggestions. Firstly, we have added "satellite-based" at the first mention of DaYu-GCP to enhance clarity for readers.

Secondly, although the original manuscript provided a detailed analysis of the characteristics of existing cloud datasets and the advantages of our product relative to each, the logical flow of the presentation was insufficient, which weakened the reading and comprehension experience for both readers and reviewers. Consequently, we have rewritten this section, integrating the comparative content directly into the paragraph describing our innovation. Our writing approach primarily involves first introducing the geostationary satellite sensors, followed by a description of the cloud products derived from these sensors. These products include the official outputs from the AGRI, AHI, ABI, and SEVIRI sensors, as well as CARE. However, due to differences in spectral channel responses among sensors and variations in the cloud retrieval algorithms employed by each, these official geostationary satellite cloud products cannot be directly merged to construct a spatiotemporally continuous global cloud product. Accordingly, the subsequent paragraph introduces existing global cloud products, including CLARA-A3, ISCCP, and SatCORPS. Nevertheless, these products either lack sufficiently fine spatial resolution or suffer from inadequate temporal resolution, making it difficult to achieve both long-term coverage and high spatiotemporal resolution simultaneously. We expect that, following these revisions, the overall quality of the manuscript has been substantially improved.

*"Satellite remote sensing is the primary means of obtaining cloud physical properties. Among*

*these platforms, geostationary satellites can continuously monitor approximately one-third of the Earth's surface day and night, providing high-frequency observations at the minute scale for long-term cloud variability studies. For example, the Advanced Geostationary Radiation Imager (AGRI) onboard the FengYun (FY)-4A/B satellites operated by the National Satellite Meteorological Centre of the China Meteorological Administration (NSMC–CMA) (Min et al., 2017; Min et al., 2020), as well as the Advanced Himawari Imager (AHI) onboard the Himawari-8/9 (H8/9) satellites operated by the Japan Aerospace Exploration Agency (JAXA), can monitor East Asia and the Pacific region (Bessho et al., 2016b, a; Iwabuchi et al., 2018). The Spinning Enhanced Visible and Infra-Red Imager (SEVIRI) onboard the Meteosat Second Generation (MSG) satellites operated by the European Organisation for the Exploitation of Meteorological Satellites (EUMETSAT) provides observations over Africa and Europe (Donny Maladji et al., 1997; Coste et al., 2017; Kocaman et al., 2022). The Geostationary Operational Environmental Satellites (GOES)-R series operated by the National Oceanic and Atmospheric Administration (NOAA), including GOES-16 to GOES-19, are equipped with the Advanced Baseline Imager (ABI) to monitor the Americas (Bin et al., 2018, 2019; Bin et al., 2020; Heidinger et al., 2020). These geostationary satellite sensors provide observations every 10–15 min, with spatial resolutions of 0.5–1 km in the visible channels and 2–5 km in the infrared channels. In contrast to geostationary satellites, polar-orbiting satellites, such as the Moderate-resolution Imaging Spectroradiometer (MODIS) onboard Aqua and Terra (Platnick et al., 2015), cannot provide high-frequency continuous observations over a given region; however, since 2000 they have offered observations with higher spatial resolution (0.25–1 km).*

*As shown in* **错误!未找到引用源。**, *based on satellite observation data, these sensors all provide official cloud physical characteristics product datasets, such as official products from the AGRI, AHI, ABI, and SEVIRI, as well as datasets from research initiatives like the Cloud Remote Sensing, Atmospheric Radiation and Renewable Energy (CARE). Most include physical characteristics such as cloud phase (CLP), cloud top height (CTH), cloud optical thickness (COT), and cloud effective radius (CER), with spatial resolutions of 2–5 km and temporal resolutions of 10–15 minutes. However, these are all regional cloud products, and most lack night-time cloud coverage. In addition, channel spectral responses differ among sensors, and the official cloud product algorithms also vary across platforms. For example, the official AHI cloud products are mainly retrieved using the Comprehensive Analysis Program for Cloud Optical Measurement (CAPCOM) multifunctional algorithm system, which integrates multi-channel threshold methods and a dual visible–near-infrared lookup table (LUT) approach. For liquid water clouds, the Mie–Lorenz scattering model is applied (Nakajima and Nakajma, 1995; Kawamoto et al., 2001), while for ice clouds, an extended Voronoi irregular ice crystal scattering model is used (Letu et al., 2020), enabling daytime cloud detection and the retrieval of COT and CER (Imai and Yoshida, 2016; Mouri et al., 2016). The ABI official products are developed by the GOES-R Algorithm Working Group, which employs LUT-based retrievals constructed from visible and near-infrared radiances during daytime, while nighttime retrievals rely on thermal infrared channels (Walther and Heidinger, 2012; Walther et al., 2013; Minnis and Heck, 2012), ultimately achieving the retrieval of CLP, CTH, COT, and CER (Pavolonis, 2010; Heidinger, 2012). These differences prevent the direct integration of official geostationary satellite cloud products into a spatiotemporally continuous global cloud product.*

*Although several global cloud physical property datasets have been developed, such as the third edition of the Satellite Application Facility on Climate Monitoring's (CM SAF) cloud, albedo,*

*and surface radiation dataset from Advanced Very High Resolution Radiometer (AVHRR) observations (CLARA-A3), which retrieves cloud amount, CTH, COT, and CER based on AVHRR measurements, its coverage is global but the temporal resolution is limited to 24 h and the spatial resolution to 0.25° (Karlsson et al., 2023b; Karlsson et al., 2023a). The International Satellite Cloud Climatology Project (ISCCP) uses AVHRR and approximately 10 km geostationary imagery to produce cloud amount, cloud types, cloud top temperature (CTT), and COT products. However, the D series (3 h; 2.5°) (Schiffer and Rossow, 1983; Rossow and Schiffer, 1991; Rossow et al., 1985) and H series (3 h; 1°) (Young et al., 2018; Rossow et al., 2022) were discontinued in 2009 and 2017, respectively. The National Oceanic and Atmospheric Administration (NOAA) Satellite ClOud and Radiation Property retrieval System (SatCORPS) employs multiple sensors, such as AHI and SEVIRI, to generate CLP, CTH, COT, and CER products, and although its temporal and spatial resolutions can reach 1 h and 3 km, respectively, the data are currently only available from 2023 onward (Trepte et al., 2019; Yost et al., 2021). These facts indicate that existing global cloud products are unable to simultaneously achieve both long temporal coverage and high spatiotemporal resolution.*

" **(In section 1 Introduction line 43-89)**

**Comment 9:**

Table 1: "specifications of" will be better than "Comparison results". Please spell out the acronyms of the datasets.

Response 9:

Thank you for your suggestions for improvement. We have now implemented the changes. The revised version is as follows:

*"Specifications of our DaYu-GCP dataset and the latest cloud physical property product dataset."*

Furthermore, we have taken note of the full names for each dataset you mentioned. In fact, we had already introduced the full names of these abbreviations in the main text preceding the tables in the manuscript. Naturally, some omissions existed in the earlier version. We have reviewed and amended the text, supplementing all abbreviations with their full names. There should now be no further omissions. You may consult the specific amendments in our newly submitted manuscript or refer to the recent 'Response 8'. As the full names have been introduced in the main text, they need not be repeated within the tables themselves.

**Comment 10:**

Line 111-112: Revise the sentence. Maybe evaluation would be "evaluate"?

Response 10:

Thank you for your comments, which are indeed correct. Here, 'evaluate' should be used to grammatically correspond with the preceding 'introduces, discusses', hence we have amended it to 'evaluates'. The revised version in the manuscript is as follows:

*"Section 2 briefly introduces data preparation and methods, while Section 3 introduces, discusses and evaluates the four main product groups: CLP, CTH, COT and CER."* **(In section 1**

**Comment 11:**

Line 161: "MOD/MYD" -> I can guess those are from collection 6, but specify the data collection version with a proper reference here first.

Response 11:

We are grateful for the reviewer's comments. Indeed, the version of the data was not clearly specified in the section you mentioned, which understandably raised questions for both reviewers and readers. We have therefore amended this section. As this is the first instance in the paper where MODIS is introduced, a more detailed explanation is warranted.

In the revised manuscript, we have clarified that the labels originate from the MODIS Level-2 cloud product (Collection 6.1), with the Terra product designated as MOD06 and the Aqua product as MYD06. Here, "06" denotes the product identification code. This clarification is now explicitly stated in the revised Section 2.1.2:

"*The MODIS instrument operates aboard two polar-orbiting satellites: Terra (launched in December 1999) and Aqua (launched in April 2002). With 36 spectral channels and a global revisit frequency of 1–2 days, MODIS's broad spectral coverage supports a wide range of applications, including vegetation-health monitoring, land-cover classification, sea-surface temperature retrieval, and cloud analysis (Hosen et al., 2023; Cai et al., 2011; Menzel et al., 2008). In this study, cloud physical properties from the MODIS Level-2 cloud product (Collection 6.1) were used as training labels, specifically CLP, CTH, CER, and COT. This product, identified by the code "06", provides data from the Terra platform (MOD06) and the Aqua platform (MYD06). Owing to its well-characterized accuracy and high data quality, the MODIS Collection 6.1 product is widely utilized as a benchmark in remote-sensing studies (Zhang et al., 2017).*" **(In section 2.1.2. Polar orbit satellite data line 135-144)**

**Comment 12:**

Line 180-181: Correct the incomplete sentence.

Response 12:

We thank the reviewers for their comments; this sentence was indeed incomplete. We have revised the sentence to place greater emphasis on the causal relationship. The amended content in the manuscript is as follows:

"*Excessively dense pressure levels in the input data may introduce unnecessary model redundancy and adversely affect training and operational efficiency. Therefore, ATP and RHP were each extracted at four identical pressure levels: 1000, 850, 500, and 300 hPa.*" **(In section 2.1.3. ERA5 data line 157-159)**

**Comment 13:**

Table 3: Correct the first row.

Response 13:

Thank you for the reviewer's suggestions. The first line provides a statement regarding the data recorded in each column of the table, and the description of the first column was not entirely appropriate. We have therefore amended this by adding 'Data matching pair'. Additionally, we have revised the names of the first column in each row: the first row is now titled 'GridSat-B1 & MOD06 Labels', the second row 'GridSat-B1 & MYD06 Labels', and the third row 'Combined MOD06 & MYD06 Labels'.

Finally, the title of Table 3 has been revised to convey the information more accurately. "Sample size and division of the matched dataset into training-set and testing-set."

| Data matching pair | Total number of samples | Training -set | Testing- set | Training-set (data augmentation) | Testing-set (data augmentation) |
|---|---|---|---|---|---|
| GridSat-B1 & MOD06 Labels | 373269 | 357744 | 15525 | 2146464 | 15525 |
| GridSat-B1 & MYD06 Labels | 324950 | 312090 | 12860 | 1872540 | 12860 |
| Combined MOD06 & MYD06 Labels | 698219 | 669834 | 28385 | 4019004 | 28385 |

**(In section 2.2 Method line 206)**

**Comment 14:**

Line 260 and Table 2: Gridsat -> GridSat

Response 14:

We are grateful to the reviewers for their meticulous scrutiny and helpful suggestions. This was indeed an oversight on our part during manuscript preparation. The correct full designation for this dataset is GridSat-B1. Consequently, all instances of 'Gridsat' in Line 260 and Table 2 of the original manuscript have been amended to 'GridSat-B1'. Furthermore, we have reviewed the entire manuscript and uniformly updated all references to 'GridSat-B1'.

**Comment 15:**

Line 299-304: Please add more details on how the initial cloud detection has been treated, which are also different between MODIS and CALIOP.

Response 15:

We are grateful for the reviewers' professional guidance. Indeed, MODIS and CALIOP exhibit fundamental differences in their initial cloud detection methods, which introduce systematic biases. The MODIS cloud mask (MOD35/MYD35), which underlies the retrieval of the subsequent MOD06/MYD06 products, employs passive sensor algorithms. In contrast, CALIOP, as an active lidar system, detects clouds by identifying significant backscatter signals that exceed background noise levels. This approach provides exceptional sensitivity to optically thin clouds and sparse

aerosol layers. MODIS, as a passive polar-orbiting satellite, offers higher accuracy than geostationary satellites and can be employed for training and verifying model precision. CALIOP, an active remote-sensing satellite, provides the highest observational accuracy and offers night-time products for assessing diurnal consistency.

This distinction was not adequately addressed in the original manuscript. Therefore, we have supplemented the text preceding paragraphs 299–304 to give readers and reviewers a clearer understanding of the differences between MODIS and CALIOP, incorporating additional references. The amended section now reads as follows:

"

*It is important to note that the initial cloud detection methodologies differ fundamentally between MODIS and CALIOP, potentially leading to systematic biases. MODIS employs a passive, multi-spectral cloud mask that uses thresholds on reflectance and brightness temperature differences (Platnick et al., 2015). CALIOP, as an active lidar, directly detects clouds by measuring laser backscatter, granting it superior sensitivity to thin clouds (Winker et al., 2010). "*

**Comment 16:**

Line 411: Missing words. Complete the sentence "Although deep learning models well trained…".

Response 16:

We thank the reviewer for pointing out this oversight. The sentence on Line 411 has been revised to complete the grammatical structure and improve clarity. We believe the revised version addresses the issue and reads more smoothly. The changes are as follows:

*"Although deep learning models are well trained, they may still misclassify inhomogeneous cloud phases in the tropics. This issue is particularly pronounced in tropical strong convective clouds with intense ice-water mixing, leading to more significant classification errors, as noted by Meyer et al. (2016)."*

**References**

Bessho, K., Date, K., Hayashi, M., Ikeda, A., Imai, T., Inoue, H., Kumagai, Y., Miyakawa, T., Murata, H., Ohno, T., Okuyama, A., Oyama, R., Sasaki, Y., Shimazu, Y., Shimoji, K., Sumida, Y., Suzuki, M., Taniguchi, H., Tsuchiyama, H., Uesawa, D., Yokota, H., and Yoshida, R.: An Introduction to Himawari-8/9— Japan's New-Generation Geostationary Meteorological Satellites, Journal of the Meteorological Society of Japan, 94, 151-183, 2016a.

Bessho, K., Date, K., Hayashi, M., Ikeda, A., Imai, T., Inoue, H., Kumagai, Y., Miyakawa, T., Murata, H., Ohno, T., Okuyama, A., Oyama, R., Sasaki, Y., Shimazu, Y., Shimoji, K., Sumida, Y., Suzuki, M., Taniguchi, H., Tsuchiyama, H., Uesawa, D., Yokota, H., and Yoshida, R.: An Introduction to Himawari-8/9— Japan’s New-Generation Geostationary Meteorological Satellites, Journal of the Meteorological Society of Japan. Ser. II, 94, 151-183, 10.2151/jmsj.2016-009, 2016b.

Bin, T., John, D., Robert, W., and Alan, R.: GOES-16 ABI navigation assessment, Proc.SPIE, 107640G, 10.1117/12.2321170,

Bin, T., John, D., Robert, W., and Alan, R.: GOES-16 and GOES-17 ABI INR assessment,

Proc.SPIE, 111271D, 10.1117/12.2529336,

Bin, T., John, J. D., Christopher, N. F., Thomas, J. G., Scott, H., Peter, J. I., Patrick, D. J., Brian, C. P., Alan, D. R., Pradeep, T., and Robert, E. W.: GOES-R series image navigation and registration performance assessment tool set, Journal of Applied Remote Sensing, 14, 032405, 10.1117/1.JRS.14.032405, 2020.

Cai, H., Zhang, S., Bu, K., Yang, J., and Chang, L.: Integrating geographical data and phenological characteristics derived from MODIS data for improving land cover mapping, Journal of Geographical Sciences, 21, 705-718, 10.1007/s11442-011-0874-1, 2011.

Coste, P., Pasternak, F., Faure, F., Jacquet, B., Bianchi, S., Donny, M. A. A., Luhmann, H. J., Hanson, C., Pili, P., and Fowler, G.: SEVIRI, the imaging radiometer on Meteosat second generation: in-orbit results and first assessment, Proc.SPIE, 105680L, 10.1117/12.2308023,

Donny Maladji, A. A., Bernard, J., and Frederick, P.: Characteristics of the Meteosat Second Generation (MSG) radiometer/imager: SEVIRI, Proc.SPIE, 19-31, 10.1117/12.298084,

Heidinger, A. K.: GOES-R Advanced Baseline Imager (ABI) Algorithm Theoretical Basis Document for Cloud Height (Version 3.0). 79 pp., https://www.star.nesdis.noaa.gov/goesr/documents/ATBDs/Baseline/ATBD_GOES-R_Cloud_Height_v3.0_Jul2012.pdf., 2012.

Heidinger, A. K., Pavolonis, M. J., Calvert, C., Hoffman, J., Nebuda, S., Straka, W., Walther, A., and Wanzong, S.: Chapter 6 - ABI Cloud Products from the GOES-R Series, in: The GOES-R Series, edited by: Goodman, S. J., Schmit, T. J., Daniels, J., and Redmon, R. J., Elsevier, 43-62, https://doi.org/10.1016/B978-0-12-814327-8.00006-8, 2020.

Helder, D., Doelling, D., Bhatt, R., Choi, T., and Barsi, J.: Calibrating Geosynchronous and Polar Orbiting Satellites: Sharing Best Practices, 10.3390/rs12172786, 2020.

Hosen, M. K., Alam, M. S., Chakraborty, T., and Golder, M. R.: Monitoring spatiotemporal and seasonal variation of agricultural drought in Bangladesh using MODIS-derived vegetation health index, Journal of Earth System Science, 132, 188, 10.1007/s12040-023-02200-3, 2023.

Huang, H., Cui, C., Cheng, L., Liu, Q., and Wang, J.: Grid interpolation algorithm based on nearest neighbor fast search, Earth Science Informatics, 5, 181-187, 10.1007/s12145-012-0106-y, 2012.

Imai, T. and Yoshida, R.: Algorithm Theoretical Basis for Himawari-8 Cloud Mask Product, Meteorological Satellite Center Technical Note, 61, 1-17, https://www.data.jma.go.jp/mscweb/technotes/msctechrep61-1.pdf., 2016.

Iwabuchi, H., Putri, N. S., Saito, M., Tokoro, Y., Sekiguchi, M., Yang, P., and Baum, B. A.: Cloud Property Retrieval from Multiband Infrared Measurements by Himawari-8, Journal of the Meteorological Society of Japan. Ser. II, 96B, 27-42, 10.2151/jmsj.2018-001, 2018.

Jiao, N., Xiang, Y., Wang, F., Zhou, G., and You, H.: Investigation of Global International GNSS Service Control Information Extraction for Geometric Calibration of Remote Sensing Images, Remote Sensing, 16, 3860, 2024.

Karlsson, K.-G., Devasthale, A., and Eliasson, S.: Global Cloudiness and Cloud Top Information from AVHRR in the 42-Year CLARA-A3 Climate Data Record Covering the Period 1979–2020, Remote Sensing, 15, 3044, 2023a.

Karlsson, K. G., Stengel, M., Meirink, J. F., Riihelä, A., Trentmann, J., Akkermans, T., Stein, D., Devasthale, A., Eliasson, S., Johansson, E., Håkansson, N., Solodovnik, I., Benas, N., Clerbaux, N., Selbach, N., Schröder, M., and Hollmann, R.: CLARA-A3: The third edition of the AVHRR-based CM SAF climate data record on clouds, radiation and surface albedo covering the period 1979 to 2023, Earth

Syst. Sci. Data, 15, 4901-4926, 10.5194/essd-15-4901-2023, 2023b.

Kawamoto, K., Nakajima, T., and Nakajima, T. Y.: A Global Determination of Cloud Microphysics with AVHRR Remote Sensing, Journal of Climate, 14, 2054-2068, https://doi.org/10.1175/1520-0442(2001)014<2054:AGDOCM>2.0.CO;2, 2001.

Kim, K.-H., Shim, P.-S., and Shin, S.: An Alternative Bilinear Interpolation Method Between Spherical Grids, Atmosphere, 10, 123, 2019.

Knapp, K. R., Ansari, S., Bain, C. L., Bourassa, M. A., Dickinson, M. J., Funk, C., Helms, C. N., Hennon, C. C., Holmes, C. D., Huffman, G. J., Kossin, J. P., Lee, H.-T., Loew, A., and Magnusdottir, G.: Globally Gridded Satellite Observations for Climate Studies, Bulletin of the American Meteorological Society, 92, 893-907, https://doi.org/10.1175/2011BAMS3039.1, 2011.

Kocaman, S., Debaecker, V., Bas, S., Saunier, S., Garcia, K., and Just, D.: A comprehensive geometric quality assessment approach for MSG SEVIRI imagery, Advances in Space Research, 69, 1462-1480, https://doi.org/10.1016/j.asr.2021.11.018, 2022.

Lee, Y., Ahn, M.-H., Kang, M., Eo, M., Kim, D., and Moon, K.-J.: Advantages of Inter-Calibration for Geostationary Satellite Sensors Onboard Twin Satellites, Geophysical Research Letters, 51, e2024GL109364, https://doi.org/10.1029/2024GL109364, 2024.

Letu, H., Yang, K., Nakajima, T. Y., Ishimoto, H., Nagao, T. M., Riedi, J., Baran, A. J., Ma, R., Wang, T., Shang, H., Khatri, P., Chen, L., Shi, C., and Shi, J.: High-resolution retrieval of cloud microphysical properties and surface solar radiation using Himawari-8/AHI next-generation geostationary satellite, Remote Sensing of Environment, 239, 111583, https://doi.org/10.1016/j.rse.2019.111583, 2020.

Li, J., Pan, B., Zhang, F., Guo, B., Li, W., Jiang, G.-M., Wu, X., and Wang, Q.: Probabilistic Retrieval of All-Day Overlapping Cloud Microphysical Properties, ADVANCES IN ATMOSPHERIC SCIENCES, 10.1007/s00376-025-5234-7, 2025.

Li, J., Zhang, F., Li, W., Tong, X., Pan, B., Li, J., Lin, H., Letu, H., and Mustafa, F.: Transfer-Learning-Based Approach to Retrieve the Cloud Properties Using Diverse Remote Sensing Datasets, IEEE Transactions on Geoscience and Remote Sensing, 61, 1-10, 10.1109/TGRS.2023.3318374, 2023a.

Li, W., Zhang, F., Lu, C., Jin, J., Shi, Y.-N., Cai, Y., Hu, S., and Han, W.: Integrated efficient radiative transfer model named Dayu for simulating the imager measurements in cloudy atmospheres, Opt. Express, 31, 15256-15288, 10.1364/OE.482762, 2023b.

Liou, K. N.: An Introduction to Atmospheric Radiation, Academic Press2002.

Liu, C., Zhang, F., Ouyang, H., Li, W., and Zhao, Z.: Diurnal Variation of Cloud Physical Properties for Tropical Cyclones Over North Atlantic in 2019–2023, Geophysical Research Letters, 52, e2025GL115566, https://doi.org/10.1029/2025GL115566, 2025.

Menzel, W. P., Frey, R. A., Zhang, H., Wylie, D. P., Moeller, C. C., Holz, R. E., Maddux, B., Baum, B. A., Strabala, K. I., and Gumley, L. E.: MODIS Global Cloud-Top Pressure and Amount Estimation: Algorithm Description and Results, Journal of Applied Meteorology and Climatology, 47, 1175-1198, https://doi.org/10.1175/2007JAMC1705.1, 2008.

Meyer, K., Yang, Y., and Platnick, S.: Uncertainties in cloud phase and optical thickness retrievals from the Earth Polychromatic Imaging Camera (EPIC), Atmos. Meas. Tech., 9, 1785-1797, 10.5194/amt-9-1785-2016, 2016.

Min, M., Li, J., Wang, F., Liu, Z., and Menzel, W. P.: Retrieval of cloud top properties from advanced geostationary satellite imager measurements based on machine learning algorithms, Remote Sensing of Environment, 239, 111616, https://doi.org/10.1016/j.rse.2019.111616, 2020.

Min, M., Wu, C., Li, C., Liu, H., Xu, N., Wu, X., Chen, L., Wang, F., Sun, F., Qin, D., Wang, X., Li,

B., Zheng, Z., Cao, G., and Dong, L.: Developing the Science Product Algorithm Testbed for Chinese Next-Generation Geostationary Meteorological Satellites: Fengyun-4 Series, Journal of Meteorological Research, 31, 708-719, 10.1007/s13351-017-6161-z, 2017.

Minnis, P. and Heck, P.: GOES-R Advanced Baseline Imager (ABI) Algorithm Theoretical Basis Document for Nighttime Cloud Optical Depth, Cloud Particle Size, Cloud Ice Water Path, and Cloud Liquid Water Path (Version 3.0), 85 pp., https://www.star.nesdis.noaa.gov/goesr/documents/ATBDs/Baseline/ATBD_GOES-R_Cloud_NCOMP_v3.0_Jul2012.pdf, 2012.

Mouri, K., Izumi, T., Suzue, H., and Yoshida, R.: Algorithm Theoretical Basis Document for Cloud Type/Phase Product, Meteorological Satellite Center Technical Note, 19-31, https://www.data.jma.go.jp/mscweb/technotes/msctechrep61-2.pdf, 2016.

Nakajima, T. and King, M. D.: Determination of the Optical Thickness and Effective Particle Radius of Clouds from Reflected Solar Radiation Measurements. Part I: Theory, Journal of Atmospheric Sciences, 47, 1878-1893, https://doi.org/10.1175/1520-0469(1990)047<1878:DOTOTA>2.0.CO;2, 1990.

Nakajima, T. Y. and Nakajma, T.: Wide-Area Determination of Cloud Microphysical Properties from NOAA AVHRR Measurements for FIRE and ASTEX Regions, Journal of Atmospheric Sciences, 52, 4043-4059, https://doi.org/10.1175/1520-0469(1995)052<4043:WADOCM>2.0.CO;2, 1995.

Pavolonis, M. J.: GOES-R Advanced Baseline Imager (ABI) Algorithm Theoretical Basis Document for Cloud Type and Cloud Phase (Version 2.0). 85 pp., https://www.star.nesdis.noaa.gov/goesr/documents/ATBDs/Baseline/ATBD_GOES-R_Cloud_Phase_Type_v2.0_Sep2010.pdf., 2010.

Platnick, S., Ackerman, S. A., King, M. D., Meyer, K., Menzel, W. P., Holz, R. E., Baum, B. A., and Yang, P.: MODIS atmosphere L2 cloud product (06_L2). NASA MODIS Adaptive Processing System, Goddard Space Flight Center [dataset], Terra (MOD06_L2): 10.5067/MODIS/MOD06_L2.006, 2015.

Rossow, W. B. and Schiffer, R. A.: ISCCP Cloud Data Products, Bulletin of the American Meteorological Society, 72, 2-20, https://doi.org/10.1175/1520-0477(1991)072<0002:ICDP>2.0.CO;2, 1991.

Rossow, W. B., Knapp, K. R., and Young, A. H.: International Satellite Cloud Climatology Project: Extending the Record, Journal of Climate, 35, 141-158, https://doi.org/10.1175/JCLI-D-21-0157.1, 2022.

Rossow, W. B., Walker, A. W., Beuschel, D. E., and Roiter, M. D.: International Satellite Cloud Climatology Project (ISCCP) Documentation of New Cloud Datasets, World Meteorological Organization, 1996.

Rossow, W. B., Mosher, F., Kinsella, E., Arking, A., Desbois, M., Harrison, E., Minnis, P., Ruprecht, E., Seze, G., Simmer, C., and Smith, E.: ISCCP Cloud Algorithm Intercomparison, Journal of Applied Meteorology and Climatology, 24, 877-903, https://doi.org/10.1175/1520-0450(1985)024<0887:ICAI>2.0.CO;2, 1985.

Schiffer, R. A. and Rossow, W. B.: The International Satellite Cloud Climatology Project (ISCCP): The First Project of the World Climate Research Programme, Bulletin of the American Meteorological Society, 64, 779-784, https://doi.org/10.1175/1520-0477-64.7.779, 1983.

Tong, X., Li, J., Zhang, F., Li, W., Pan, B., Li, J., and Letu, H.: The Deep-Learning-Based Fast Efficient Nighttime Retrieval of Thermodynamic Phase From Himawari-8 AHI Measurements, Geophysical Research Letters, 50, e2022GL100901, https://doi.org/10.1029/2022GL100901, 2023.

Trepte, Q. Z., Minnis, P., Sun-Mack, S., Yost, C. R., Chen, Y., Jin, Z., Hong, G., Chang, F. L., Smith,

W. L., Bedka, K. M., and Chee, T. L.: Global Cloud Detection for CERES Edition 4 Using Terra and Aqua MODIS Data, IEEE Transactions on Geoscience and Remote Sensing, 57, 9410-9449, 10.1109/TGRS.2019.2926620, 2019.

Walther, A. and Heidinger, A. K.: Implementation of the Daytime Cloud Optical and Microphysical Properties Algorithm (DCOMP) in PATMOS-x, Journal of Applied Meteorology and Climatology, 51, 1371-1390, https://doi.org/10.1175/JAMC-D-11-0108.1, 2012.

Walther, A., Straka, W., and Heidinger, A. K.: GOES-R Advanced Baseline Imager (ABI) Algorithm Theoretical Basis Document for Daytime Cloud Optical and Microphysical Properties (DCOMP) (Version 3.0), 66 pp., https://www.star.nesdis.noaa.gov/goesr/documents/ATBDs/Baseline/ATBD_GOES-R_Cloud_DCOMP_v3.0_Jun2013.pdf, 2013.

Winker, D., Pelon, J., Coakley, J., Ackerman, S., Charlson, R., Colarco, P., Flamant, P., Fu, Q., Hoff, R., Kittaka, C., Kubar, T., Treut, H., McCormick, M., Mégie, G., Poole, L., Trepte, C., Vaughan, M., and Wielicki, B.: The CALIPSO Mission: A Global 3D View of Aerosols and Clouds, Bulletin of the American Meteorological Society, 91, 1211-1230, 10.1175/2010BAMS3009.1, 2010.

Xiao, H., Zhang, F., Wang, L., Pan, B., Zhu, Y., Wang, M., Li, W., Guo, B., and Li, J.: High-resolution ensemble retrieval of cloud properties for all-day based on geostationary satellite, npj Climate and Atmospheric Science, 8, 386, 10.1038/s41612-025-01263-x, 2025.

Yost, C. R., Minnis, P., Sun-Mack, S., Chen, Y., and Smith, W. L.: CERES MODIS Cloud Product Retrievals for Edition 4—Part II: Comparisons to CloudSat and CALIPSO, IEEE Transactions on Geoscience and Remote Sensing, 59, 3695-3724, 10.1109/TGRS.2020.3015155, 2021.

Young, A. H., Knapp, K. R., Inamdar, A., Hankins, W., and Rossow, W. B.: The International Satellite Cloud Climatology Project H-Series climate data record product, Earth Syst. Sci. Data, 10, 583-593, 10.5194/essd-10-583-2018, 2018.

Zhang, Z., Dong, X., Xi, B., Song, H., Ma, P.-L., Ghan, S. J., Platnick, S., and Minnis, P.: Intercomparisons of marine boundary layer cloud properties from the ARM CAP-MBL campaign and two MODIS cloud products, Journal of Geophysical Research: Atmospheres, 122, 2351-2365, https://doi.org/10.1002/2016JD025763, 2017.

Zhao, Z., Zhang, F., Li, W., and Li, J.: Image-Based Retrieval of All-Day Cloud Physical Parameters for FY4A/AGRI and Its Application Over the Tibetan Plateau, Journal of Geophysical Research: Atmospheres, 129, e2024JD041032, https://doi.org/10.1029/2024JD041032, 2024a.

Zhao, Z., Zhang, F., Wu, Q., Li, Z., Tong, X., and Li, J.: Cloud Identification and Properties Retrieval of the Fengyun-4A Satellite Using a ResUnet Model, IEEE Transactions on Geoscience and Remote Sensing, 61, 1-18, 10.1109/TGRS.2023.3252023, 2023.

Zhao, Z., Zhang, F., Li, W., Meteorology, J. L. C.-F. J. L. o. M., Atmospheric, D. o., Sciences, O., Sciences, I. o. S., University, F., China, Waves, K. L. f. I. S. o. E., Education, M. o., Science, S. o. M., and Technology: Image-Based Retrieval of All-Day Cloud Physical Parameters for FY4A/AGRI and Its Application Over the Tibetan Plateau, Journal of Geophysical Research: Atmospheres, 129, 2024b.